# Stochastic Newton Proximal Extragradient Method

**Ruichen Jiang**
ECE Department
UT Austin
rjiang@utexas.edu

**Michał Dereziński**
EECS Department
University of Michigan
derezin@umich.edu

**Aryan Mokhtari**
ECE Department
UT Austin
mokhtari@austin.utexas.edu

## Abstract

Stochastic second-order methods achieve fast local convergence in strongly convex optimization by using noisy Hessian estimates to precondition the gradient. However, these methods typically reach superlinear convergence only when the stochastic Hessian noise diminishes, increasing per-iteration costs over time. Recent work in [1] addressed this with a Hessian averaging scheme that achieves superlinear convergence without higher per-iteration costs. Nonetheless, the method has slow global convergence, requiring up to $\tilde{\mathcal{O}}(\kappa^2)$ iterations to reach the superlinear rate of $\tilde{\mathcal{O}}((1/t)^{t/2})$, where $\kappa$ is the problem's condition number. In this paper, we propose a novel stochastic Newton proximal extragradient method that improves these bounds, achieving a faster global linear rate and reaching the same fast superlinear rate in $\tilde{\mathcal{O}}(\kappa)$ iterations. We accomplish this by extending the Hybrid Proximal Extragradient (HPE) framework, achieving fast global and local convergence rates for strongly convex functions with access to a noisy Hessian oracle.

## 1 Introduction

In this paper, we focus on the use of second-order methods for solving the optimization problem

$$\min_{\mathbf{x} \in \mathbb{R}^d} f(\mathbf{x}), \tag{1}$$

where $f : \mathbb{R}^d \to \mathbb{R}$ is strongly convex and twice differentiable. There is an extensive literature on second-order methods and their fast local convergence properties; e.g., [2–5]. However, these results necessitate access to the exact Hessian, which can pose computational challenges. To address this issue, several studies have explored scenarios where only the exact gradient can be queried, while a stochastic estimate of the Hessian is available—similar to the setting we investigate in this paper. This oracle model is commonly encountered in large-scale machine learning problems, as computing the gradient is often much less expensive than computing the Hessian, and approximating the Hessian is a more affordable approach. Specifically, consider a finite-sum minimization problem $\min_{x \in \mathbb{R}^d} \sum_{i=1}^n f_i(x)$, where $n$ denotes the number of data points and $d$ denotes the dimension of the problem. To achieve a fast convergence rate, standard first-order methods need to compute one full gradient in each iteration, resulting in a per-iteration computational cost of $\mathcal{O}(nd)$. In contrast, implementing a second-order method such as damped Newton's method involves computing the full Hessian, which costs $\mathcal{O}(nd^2)$. An inexact Hessian estimate can be constructed efficiently at a cost of $\mathcal{O}(sd^2)$, where $s$ is the sketch size or subsampling size [1, 6]. Hence, when the number of samples $n$ significantly exceeds $d$, the per-iteration cost of stochastic second-order methods becomes comparable to that of first-order methods. Moreover, using second-order information often reduces the number of iterations needed to converge, thereby lowering overall computational complexity.

A common template among stochastic second-order methods is to combine a deterministic second-order method, such as Newton's method or cubic regularized Newton method, with techniques such as Hessian subsampling [7–12] or Hessian sketching [4, 6, 13] that only require a noisy estimate

38th Conference on Neural Information Processing Systems (NeurIPS 2024).

Table 1: Comparison between Algorithm 1 and the stochastic Newton method in [1], in terms of how many iterations it takes to transition to each phase, and the convergence rates achieved. We drop constant factors as well as logarithmic dependence and $1/\delta$, and assume $1/\text{poly}(\kappa) \leq \Upsilon \leq \mathcal{O}(\kappa)$.

| Methods | Weights | Linear phase | | Initial superlinear phase | | Final superlinear phase | |
|---|---|---|---|---|---|---|---|
| | | $\mathcal{T}_1$ | rate $\phi$ | $\mathcal{T}_2$ | rate $\theta_t^{(1)}$ | $\mathcal{T}_3$ | rate $\theta_t^{(2)}$ |
| Stochastic Newton [1] | Uniform | $\Upsilon^2$ | $1-\kappa^{-2}$ | $\kappa^3$ | $\frac{\kappa^3}{t}$ | $\frac{\kappa^6}{\Upsilon^2}$ | $\Upsilon\sqrt{\frac{\log(t)}{t}}$ |
| | Non-uniform | $\Upsilon^2$ | $1-\kappa^{-2}$ | $\kappa^2$ | $\frac{\kappa^{4\log(\kappa)+1}}{t^{\log(t)}}$ | $\kappa^2$ | $\Upsilon\frac{\log(t)}{\sqrt{t}}$ |
| Stochastic NPE (**Ours**) | Uniform | $\frac{\Upsilon^2}{\kappa^2}$ | $1-\kappa^{-1}$ | $\kappa^2$ | $\frac{\kappa^2}{t}$ | $\frac{\kappa^4}{\Upsilon^2}$ | $\Upsilon\sqrt{\frac{\log(t)}{t}}$ |
| | Non-uniform | $\frac{\Upsilon^2}{\kappa^2}$ | $1-\kappa^{-1}$ | $\Upsilon^2+\kappa$ | $\frac{(\Upsilon^2+\kappa)^{\log(\Upsilon^2+\kappa)+1}}{t^{\log(t)}}$ | $\Upsilon^2+\kappa$ | $\Upsilon\frac{\log(t)}{\sqrt{t}}$ |

of the Hessian. We refer the reader to [14, 15] for recent surveys and empirical comparisons. In terms of convergence guarantees, the majority of these works, including [4, 6, 8–11, 13], have shown that stochastic second-order methods exhibit a global linear convergence and a local linear-quadratic convergence, either with high probability or in expectation. The linear-quadratic behavior holds when

$$\|\mathbf{x}_{t+1} - \mathbf{x}^*\| \leq c_1\|\mathbf{x}_t - \mathbf{x}^*\| + c_2\|\mathbf{x}_t - \mathbf{x}^*\|^2, \tag{2}$$

where $\mathbf{x}^*$ denotes the optimal solution of Problem (1) and $c_1, c_2$ are constants depending on the sample/sketch size at each step. In particular, the presence of the linear term in (2) implies that the algorithm can only achieve linear convergence when the iterate is sufficiently close to the optimal solution $\mathbf{x}^*$. Consequently, as discussed in [9, 10], to achieve superlinear convergence, the coefficient $c_1 = c_{1,t}$ needs to gradually decrease to zero as $t$ increases. However, since $c_1$ is determined by the magnitude of the stochastic noise in the Hessian estimate, this in turn demands the sample/sketch size to increase across the iterations, leading to a blow-up of the per-iteration computational cost.

The only prior work addressing this limitation and achieving a superlinear rate for a stochastic second-order method without requiring the stochastic Hessian noise to converge to zero is by [1]. It uses a weighted average of all past Hessian approximations as the current Hessian estimate. This approach reduces stochastic noise variance in the Hessian estimate, though it introduces bias to the Hessian approximation matrix. When combined with Newton's method, it was shown that the proposed method achieves local superlinear convergence with a non-asymptotic rate of $(\Upsilon\sqrt{\log(t)/t})^t$ with high probability, where $\Upsilon$ characterizes the noise level of the stochastic Hessian oracle (see Assumption 4). However, the method may require many iterations to achieve superlinear convergence. Specifically, with the uniform averaging scheme, it takes $\tilde{\mathcal{O}}(\kappa^3)$ iterations before the method starts converging superlinearly and $\tilde{\mathcal{O}}(\kappa^6/\Upsilon^2)$ iterations before it reaches the final superlinear rate. Here, $\kappa = L_1/\mu$ denotes the condition number of the function $f$, where $L_1$ is the Lipschitz constant of the gradient and $\mu$ is the strong convexity parameter. To address this, [1] proposed a weighted averaging scheme that assigns more weight to recent Hessian estimates, improving both transition points to $\tilde{\mathcal{O}}(\Upsilon^2 + \kappa^2)$ while achieving a slightly slower superlinear rate of $\mathcal{O}(\Upsilon\log(t)/\sqrt{t})$.

**Our contributions.** In this paper, we improve the complexity of Stochastic Newton in [1] with a method that attains a superlinear rate in significantly fewer iterations. As shown in Table 1, our method requires fewer iterations for linear convergence, denoted as $\mathcal{T}_1$, by a factor of $\kappa^2$ compared to [1]. Additionally, our method achieves a linear convergence rate of $(1 - \mathcal{O}(1/\kappa))^t$, outperforming the $(1 - \mathcal{O}(1/\kappa^2))^t$ rate in [1]. Thus, our method reaches the local neighborhood of the optimal solution $\mathbf{x}^*$ and transitions from linear to superlinear convergence faster. Specifically, the second transition point, $\mathcal{T}_2$, is smaller by a factor of $\kappa$ in both uniform and non-uniform averaging schemes when $\Upsilon = \mathcal{O}(\sqrt{\kappa})$. Similarly, our method's initial superlinear rate has a better dependence on $\kappa$, leading to fewer iterations, $\mathcal{T}_3$, to enter the final superlinear phase. To achieve this result, we use the hybrid proximal extragradient (HPE) framework [16, 17] instead of Newton's method as the base algorithm. The HPE framework provides a principled approach for designing second-order methods with superior global convergence guarantees [3, 17–20]. However, [16] and subsequent works focus on cases where $f$ is merely convex, not leveraging strong convexity. Thus, we modify the HPE framework to suit our setting. Specifically, we relax the error condition for computing the proximal step in HPE, enabling a larger step size when the iterate is close to the optimal solution, crucial for achieving the final superlinear convergence rate.

## 2 Preliminaries

In this section, we formally present our assumptions.

**Assumption 1.** *The function $f$ is twice differentiable and $\mu$-strongly convex.*

**Assumption 2.** *The Hessian $\nabla^2 f$ satisfies $\|\nabla^2 f(\mathbf{x}) - \nabla^2 f(\mathbf{y})\| \leq M_1$.*

**Assumption 3.** *The Hessian $\nabla^2 f$ is $L_2$-Lipschitz, i.e., $\|\nabla^2 f(\mathbf{x}) - \nabla^2 f(\mathbf{y})\| \leq L_2 \|\mathbf{x} - \mathbf{y}\|_2$.*

Assumption 2 is more general than the assumption that $\nabla f$ is $L_1$-Lipschitz. In particular, if the latter assumption holds, then $M_1 \leq L_1$. Moreover, we define $\kappa \triangleq M_1/\mu$ as the condition number.

To simplify our notation, we denote the exact gradient $\nabla f(\mathbf{x})$ and the exact Hessian $\nabla^2 f(\mathbf{x})$ of the objective function by $\mathbf{g}(\mathbf{x})$ and $\mathbf{H}(\mathbf{x})$, respectively. As mentioned earlier, we assume that we have access to the exact gradient, but we only have access to a noisy estimate of the Hessian denoted by $\hat{\mathbf{H}}(\mathbf{x})$. In fact, we require a mild assumption on the Hessian noise. We define the stochastic Hessian noise as $\mathbf{E}(\mathbf{x}) \triangleq \hat{\mathbf{H}}(\mathbf{x}) - \mathbf{H}(\mathbf{x})$, where it is assumed to be mean zero and sub-exponential.

**Assumption 4.** *If we define $\mathbf{E}(\mathbf{x}) \triangleq \hat{\mathbf{H}}(\mathbf{x}) - \mathbf{H}(\mathbf{x})$, then $\mathbb{E}[\mathbf{E}(\mathbf{x})] = 0$ and $\mathbb{E}[\|\mathbf{E}(\mathbf{x})\|^p] \leq p! \Upsilon_E^p / 2$ for all integers $p \geq 2$. Also, define $\Upsilon \triangleq \Upsilon_E / \mu$ to be the relative noise level.*

**Assumption 5.** *The Hessian approximation matrix is positive semi-definite, i.e., $\hat{\mathbf{H}}(\mathbf{x}) \succeq 0$, $\forall \mathbf{x} \in \mathbb{R}^d$.*

**Stochastic Hessian construction.** The two most popular approaches to construct stochastic Hessian approximations are "subsampling" and "sketching". *Hessian subsampling* is designed for a finite-sum objective of the form $f(\mathbf{x}) = \frac{1}{n} \sum_{i=1}^{n} f_i(\mathbf{x})$, where $n$ is the number of samples. In each iteration, a subset $S \subset \{1, 2, \ldots, n\}$ is drawn uniformly at random, and then the subsampled Hessian at $\mathbf{x}$ is constructed as $\hat{\mathbf{H}}(\mathbf{x}) = \frac{1}{|S|} \sum_{i \in S} \nabla^2 f_i(\mathbf{x})$. In this case, if each $f_i$ is convex, then the condition in Assumption 5 is satisfied. Moreover, if we further assume that $\|\nabla^2 f_i(\mathbf{x})\| \leq c M_1$ for some $c > 0$ and for all $i$, then Assumption 4 is satisfied with $\Upsilon = \mathcal{O}(\sqrt{c\kappa \log(d)/|S|} + c\kappa \log(d)/|S|)$ (see [1, Example 1]). The other approach is *Hessian sketching*, applicable when the Hessian $\mathbf{H}$ can be easily factorized as $\mathbf{H} = \mathbf{M}^\top \mathbf{M}$, where $\mathbf{M} \in \mathbb{R}^{n \times d}$ is the square-root Hessian matrix, and $n$ is the number of samples. This is the case for generalized linear models; see [1]. To form the sketched Hessian, we draw a random sketch matrix $\mathbf{S} \in \mathbb{R}^{s \times n}$ with sketch size $s$ from a distribution $\mathcal{D}$ that satisfies $\mathbb{E}_{\mathcal{D}}[\mathbf{S}^\top \mathbf{S}] = \mathbf{I}$. The sketched Hessian is then $\hat{\mathbf{H}} = \mathbf{M}^\top \mathbf{S}^\top \mathbf{S} \mathbf{M}$. In this case, Assumption 5 is automatically satisfied. Moreover, for Gaussian sketch, Assumption 4 is satisfied with $\Upsilon = \mathcal{O}(\kappa(\sqrt{d/s} + d/s))$ (see [1, Example 2]).

*Remark* 1. The above assumptions are common in the study of stochastic second-order methods, appearing in works on Subsampled Newton [7–10, 12], Newton Sketch [4, 6, 13], and notably, [1]. The strong convexity requirement is crucial as stochastic second-order methods have a clear advantage over first-order methods like gradient descent when the function is strongly convex. Specifically, stochastic second-order methods attain a superlinear convergence rate, as shown in this paper, which is superior to the linear rate of first-order methods.

## 3 Stochastic Newton Proximal Extragradient

Our approach involves developing a stochastic Newton-type method grounded in the Hybrid Proximal Extragradient (HPE) framework and its second-order variant. Therefore, before introducing our proposed algorithm, we will provide a brief overview of the core principles of the HPE framework. Following this, we will present our method as it applies to the specific setting addressed in this paper.

**Hybrid Proximal Extragradient.** Next, we first present the Hybrid Proximal Extragradient (HPE) framework for strongly convex functions. To solve problem (1), the HPE algorithm consists of two steps. In the first step, given $\mathbf{x}_t$, we find a mid-point $\hat{\mathbf{x}}_t$ by applying an inexact proximal point update $\hat{\mathbf{x}}_t \approx \mathbf{x}_t - \eta_t \nabla f(\hat{\mathbf{x}}_t)$, where $\eta_t$ is the step size. More precisely, we require

$$\|\hat{\mathbf{x}}_t - \mathbf{x}_t + \eta_t \nabla f(\hat{\mathbf{x}}_t)\| \leq \alpha \sqrt{\gamma_t} \|\hat{\mathbf{x}}_t - \mathbf{x}_t\|, \tag{3}$$

where $\gamma_t = 1 + 2\eta_t \mu$, $\mu$ is the strong convexity parameter, and $\alpha \in (0, 1)$ is a user-specified parameter. Then, in the second step, we perform the extra-gradient update and compute $\mathbf{x}_{t+1}$ based on

$$\mathbf{x}_{t+1} = \frac{1}{\gamma_t} (\mathbf{x}_t - \eta_t \nabla f(\hat{\mathbf{x}}_t)) + \left(1 - \frac{1}{\gamma_t}\right) \hat{\mathbf{x}}_t, \tag{4}$$

The weights $\frac{1}{\gamma_t}$ in the above convex combination are chosen to optimize the convergence rate.

*Remark* 2. When $\mu = 0$, the algorithm outline above reduces to the original HPE framework studied in [16, 18]. Our modification in (3) is inspired by [21] and allows a larger error when performing the inexact proximal point update, which turns out to be crucial for achieving a fast superlinear convergence rate. Moreover, the modification in (4) has been adopted in [22].

**Stochastic Newton Proximal Extragradient (SNPE).** The HPE method described above provides a useful algorithmic framework, instead of a directly implementable method. The main challenge comes from implementing the first step in (3), which involves an inexact proximal point update. Specifically, the naive approach is to solve the *implicit nonlinear equation* $\mathbf{x} - \mathbf{x}_t + \eta_t \nabla f(\mathbf{x}) = 0$, which can be as costly as solving the original problem in (1). To address this issue, [18] proposed to approximate the gradient operator $\nabla f(\mathbf{x})$ by its local linearization $\nabla f(\mathbf{x}_t) + \nabla^2 f(\mathbf{x}_t)(\mathbf{x} - \mathbf{x}_t)$, and then compute $\hat{\mathbf{x}}_t$ by solving the linear system of equations $\hat{\mathbf{x}}_t - \mathbf{x}_t + \eta_t(\nabla f(\mathbf{x}_t) + \nabla^2 f(\mathbf{x}_t)(\hat{\mathbf{x}}_t - \mathbf{x}_t)) = 0$. This leads to the Newton proximal extragradient method that was proposed and analyzed in [18].

However, in our setting, the exact Hessian $\nabla^2 f(\mathbf{x}_t)$ is not available. Thus, we construct a stochastic Hessian approximation $\tilde{\mathbf{H}}_t$ from our noisy Hessian oracle as a surrogate of $\nabla^2 f(\mathbf{x}_t)$. We will elaborate on the construction of $\tilde{\mathbf{H}}_t$ later, but for the present discussion assume that this stochastic Hessian approximation $\tilde{\mathbf{H}}_t$ is already provided. Then in the first step, we will compute $\hat{\mathbf{x}}_t$ by

$$\hat{\mathbf{x}}_t = \mathbf{x}_t - \eta_t(\nabla f(\mathbf{x}_t) + \tilde{\mathbf{H}}_t(\hat{\mathbf{x}}_t - \mathbf{x}_t)), \tag{5}$$

where we replace $\nabla f(\hat{\mathbf{x}}_t)$ by its local linear approximation $\nabla f(\mathbf{x}_t) + \tilde{\mathbf{H}}_t(\hat{\mathbf{x}}_t - \mathbf{x}_t)$. Moreover, (5) is equivalent to solving the following linear system of equations $(\mathbf{I} + \eta_t \tilde{\mathbf{H}}_t)(\mathbf{x} - \mathbf{x}_t) = -\eta_t \nabla f(\mathbf{x}_t)$. For ease of presentation, we set $\hat{\mathbf{x}}_t$ as the exact solution of this system, leading to

$$\hat{\mathbf{x}}_t = \mathbf{x}_t - \eta_t(\mathbf{I} + \eta_t \tilde{\mathbf{H}}_t)^{-1} \nabla f(\mathbf{x}_t). \tag{6}$$

However, we note that an inexact solution to this linear system is also sufficient for our convergence guarantees so long as $\|(\mathbf{I} + \eta_t \tilde{\mathbf{H}}_t)(\hat{\mathbf{x}}_t - \mathbf{x}_t) + \eta_t \nabla f(\mathbf{x}_t)\| \le \frac{\alpha}{2}\|\hat{\mathbf{x}}_t - \mathbf{x}_t\|$; We refer the reader to Appendix A.3 for details. Additionally, since we employed a linear approximation to determine the mid-point $\hat{\mathbf{x}}_t$, the condition in (3) may no longer be satisfied. Consequently, it is crucial to verify the accuracy of our approximation after selecting $\hat{\mathbf{x}}_t$. To achieve this, we implement a line-search scheme to ensure that the step size is not large and the linear approximation error is small.

Next, we discuss constructing the stochastic Hessian approximation $\tilde{\mathbf{H}}_t$. A simple strategy is using $\hat{\mathbf{H}}(\mathbf{x}_t)$ instead of $\nabla^2 f(\mathbf{x}_t)$, but the Hessian noise would lead to a highly inaccurate approximation of the prox operator, ruining the superlinear convergence rate. To reduce Hessian noise, we follow [1] and use an averaged Hessian estimate $\tilde{\mathbf{H}}(\mathbf{x}_t)$. We consider two schemes: (i) uniform averaging; (ii) non-uniform averaging with general weights. In the first case, $\tilde{\mathbf{H}}_t = \frac{1}{t+1} \sum_{i=0}^{t} \hat{\mathbf{H}}(\mathbf{x}_t)$ uniformly averages past stochastic Hessian approximations. Motivated by the central limit theorem for martingale differences, we expect $\tilde{\mathbf{H}}_t$ to have smaller variance than $\hat{\mathbf{H}}(\mathbf{x}_t)$. It can be implemented online as $\tilde{\mathbf{H}}_t = \frac{t}{t+1}\hat{\mathbf{H}}_{t-1} + \frac{1}{t+1}\hat{\mathbf{H}}(\mathbf{x}_t)$, without storing past Hessian estimates. However, $\hat{\mathbf{H}}(\mathbf{x}_t)$ is a *biased* estimator of $\nabla^2 f(\mathbf{x}_t)$, since it incorporates stale Hessian information. To address the bias-variance trade-off, the second case uses non-uniform averaging to weight recent Hessian estimates more. Given an increasing non-negative weight sequence $\{w_t\}_{t=-1}^{\infty}$ with $w_{-1} = 0$, the running average is:

$$\tilde{\mathbf{H}}_t = \frac{w_{t-1}}{w_t}\tilde{\mathbf{H}}_{t-1} + \left(1 - \frac{w_{t-1}}{w_t}\right)\hat{\mathbf{H}}(\mathbf{x}_t). \tag{7}$$

Equivalently, with $z_{i,t} = \frac{w_i - w_{i-1}}{w_t}$, $\tilde{\mathbf{H}}_t$ can be written as $\sum_{i=0}^{t} z_{i,t}\hat{\mathbf{H}}(\mathbf{x}_i)$. We discuss uniform averaging in Section 4 and non-uniform averaging in Section 5.

Building on the discussion thus far, we are ready to integrate all the components and present our Stochastic Newton Proximal Extragradient (SNPE) method. The steps of SNPE are summarized in Algorithm 1. Each iteration of our SNPE method includes two stages. In the first stage, starting with the current point $\mathbf{x}_t$, we first query the noisy Hessian oracle and compute the averaged stochastic Hessian $\tilde{\mathbf{H}}_t$ from (7), as stated in Step 4. Then given the gradient $\nabla f(\mathbf{x}_t)$, the Hessian approximation $\tilde{\mathbf{H}}_t$, and an initial trial step size $\sigma_t$, we employ a backtracking line search to obtain $\eta_t$ and $\hat{\mathbf{x}}_t$, as stated in Step 6 of Algorithm 1. Specifically, in this step, we set $\eta_t \leftarrow \sigma_t$ and compute $\hat{\mathbf{x}}_t$ as suggested

| **Algorithm 1** Stochastic NPE | **Subroutine 1** $(\eta, \hat{\mathbf{x}}) = \mathrm{BLS}(\mathbf{x}, \mathbf{g}, \tilde{\mathbf{H}}, \alpha, \beta, \sigma)$ |
|---|---|

**Algorithm 1** Stochastic NPE

1: **Input:** $\mathbf{x}_0 \in \mathbb{R}^d$, weights $\{w_t\}_{t=0}^{\infty}$, line-search parameters $\alpha, \beta \in (0,1)$, initial step size $\sigma_0 > 0$
2: **Initialize:** $\tilde{\mathbf{H}}_{-1} = \mathbf{0}$ and $w_{-1} = 0$
3: **for** $t = 0, 1, \dots$ **do**
4:     Obtain a stochastic Hessian $\hat{\mathbf{H}}_t = \hat{\mathbf{H}}(\mathbf{x}_t)$
5:     Compute $\tilde{\mathbf{H}}_t = \frac{w_{t-1}}{w_t}\tilde{\mathbf{H}}_{t-1} + (1 - \frac{w_{t-1}}{w_t})\hat{\mathbf{H}}_t$
6:     $(\eta_t, \hat{\mathbf{x}}_t) = \mathrm{BLS}(\mathbf{x}_t, \nabla f(\mathbf{x}_t), \tilde{\mathbf{H}}_t, \alpha, \beta, \sigma_t)$
7:     Let $\gamma_t = 1 + 2\eta_t\mu$ and compute
       $\mathbf{x}_{t+1} = \frac{1}{\gamma_t}(\mathbf{x}_t - \eta_t\nabla f(\hat{\mathbf{x}}_t)) + (1 - \frac{1}{\gamma_t})\hat{\mathbf{x}}_t$
8:     Set $\sigma_{t+1} = \eta_t/\beta$
9: **end for**

**Subroutine 1** $(\eta, \hat{\mathbf{x}}) = \mathrm{BLS}(\mathbf{x}, \mathbf{g}, \tilde{\mathbf{H}}, \alpha, \beta, \sigma)$

1: **Input:** current iterate $\mathbf{x} \in \mathbb{R}^d$, gradient $\mathbf{g} \in \mathbb{R}^d$, Hessian approximation $\tilde{\mathbf{H}} \in \mathbb{R}^{d \times d}$, line-search parameters $\alpha, \beta \in (0,1)$, initial trial step size $\sigma > 0$
2: Set $\eta \leftarrow \sigma$ and $\hat{\mathbf{x}} \leftarrow \mathbf{x} - \eta(\mathbf{I} + \eta\tilde{\mathbf{H}})^{-1}\mathbf{g}$
3: Set $\gamma \leftarrow 1 + 2\eta\mu$
4: **while** $\|\hat{\mathbf{x}} - \mathbf{x} + \eta\nabla f(\hat{\mathbf{x}})\| > \alpha\sqrt{\gamma}\|\hat{\mathbf{x}} - \mathbf{x}\|$ **do**
5:     Set $\eta \leftarrow \beta\eta$ and $\hat{\mathbf{x}} \leftarrow \mathbf{x} - \eta(\mathbf{I} + \eta\tilde{\mathbf{H}})^{-1}\mathbf{g}$
6:     Set $\gamma \leftarrow 1 + 2\eta\mu$
7: **end while**
8: **Output:** $\eta$ and $\hat{\mathbf{x}}$

in (6). If $\hat{\mathbf{x}}_t$ and its corresponding step size $\eta_t$ satisfy (3), meaning the linear approximation error is small, then the step size $\eta_t$ and the mid-point $\hat{\mathbf{x}}_t$ are accepted and we proceed to the second stage of SNPE. If not, we backtrack the step size $\eta_t$ and try a smaller step size $\beta\eta_t$, where $\beta \in (0,1)$ is a user-specified parameter. We repeat the process until the condition in (3) is satisfied. The details of the backtracking line search scheme are summarized in Subroutine 1. After completing the first stage and obtaining the pair $(\eta_t, \hat{\mathbf{x}}_t)$, we proceed to the extragradient step and follow the update in (4), as in Step 7 of Algorithm 1. Finally, before moving to the next time index, we follow a warm-start strategy and set the next initial trial step size $\sigma_{t+1}$ as $\eta_t/\beta$, as shown in Step 8 of Algorithm 1.

*Remark* 3. Similar to the analysis in [22], we can show that the total number of line search steps after $t$ iterations can be bounded by $2t - 1 + \log(\frac{\sigma_0}{\eta_{t-1}})$. Moreover, when $t$ is large enough, on average the line search requires 2 steps per iteration. We defer the details to Appendix A.4.

*Remark* 4. Our motivation behind the choice $\sigma_{t+1} = \eta_t/\beta$ is to allow the step size to grow, which is necessary for achieving a superlinear convergence rate. Specifically, as shown in Proposition 1 below, we require the step size $\eta_t$ to go to infinity to ensure that $\lim_{t\to\infty} \frac{\|\mathbf{x}_{t+1}-\mathbf{x}^*\|}{\|\mathbf{x}_t-\mathbf{x}^*\|} = 0$. Note that this would not be possible if we simply set $\sigma_{t+1} = \eta_t$, since it would automatically result in $\eta_{t+1} \le \sigma_{t+1} \le \eta_t$. Moreover, this condition $\sigma_{t+1} = \eta_t/\beta$ is explicitly utilized in Lemmas 8 and 16 in the Appendix, where we demonstrate that $\eta_t$ can be lower bounded by the minimum of $\sigma_0/\beta^t$ and another term. We should also note that this more aggressive choice of the initial step size at each round could potentially increase the number of backtracking steps. However, as mentioned above, this does not cause a significant issue, since the average number of backtracking steps per iteration can be bounded by a constant close to 2.

### 3.1 Key properties of SNPE

This section outlines key properties of SNPE, applied in Sections 4 and 5 to determine its convergence rates. The first result reveals the connection between SNPE's convergence rate and the step size $\eta_t$.

**Proposition 1.** *Let $\{\mathbf{x}_t\}_{t\ge 0}$ and $\{\hat{\mathbf{x}}_t\}_{t\ge 0}$ be the iterates generated by Algorithm 1. Then for any $t \ge 0$, we have $\|\mathbf{x}_{t+1} - \mathbf{x}^*\|^2 \le \|\mathbf{x}_t - \mathbf{x}^*\|^2(1 + 2\eta_t\mu)^{-1}$.*

Proposition 1 guarantees that the distance to the optimal solution is monotonically decreasing, and it shows a larger step size implies faster convergence. Hence, we need to provide an explicit lower bound on the step size. This task is accomplished in the next lemma. For ease of notation, we let $\mathcal{B}$ be the set of iteration indices where the line search scheme backtracks, i.e., $\mathcal{B} \triangleq \{t : \eta_t < \sigma_t\}$. Moreover, we use $\mathbf{g}(\mathbf{x})$ and $\mathbf{H}(\mathbf{x})$ to denote the gradient $\nabla f(\mathbf{x})$ and the Hessian $\nabla^2 f(\mathbf{x})$, respectively.

**Lemma 1.** *For $t \notin \mathcal{B}$, we have $\eta_t = \sigma_t$. For $t \in \mathcal{B}$, let $\tilde{\eta}_t = \eta_t/\beta$ and $\tilde{\mathbf{x}}_t = \mathbf{x}_t - \tilde{\eta}_t(\mathbf{I} + \tilde{\eta}_t\tilde{\mathbf{H}}_t)^{-1}\nabla f(\mathbf{x}_t)$. Then, $\|\tilde{\mathbf{x}}_t - \mathbf{x}_t\| \le \frac{1}{\beta}\|\hat{\mathbf{x}}_t - \mathbf{x}_t\|$. Moreover,*

$$\eta_t \ge \max\left\{ \frac{\alpha\beta\|\tilde{\mathbf{x}}_t - \mathbf{x}_t\|}{\|\mathbf{g}(\tilde{\mathbf{x}}_t) - \mathbf{g}(\mathbf{x}_t) - \tilde{\mathbf{H}}_t(\tilde{\mathbf{x}}_t - \mathbf{x}_t)\|}, \frac{2\alpha^2\beta\mu\|\tilde{\mathbf{x}}_t - \mathbf{x}_t\|^2}{\|\mathbf{g}(\tilde{\mathbf{x}}_t) - \mathbf{g}(\mathbf{x}_t) - \tilde{\mathbf{H}}_t(\tilde{\mathbf{x}}_t - \mathbf{x}_t)\|^2} \right\}.$$

As Lemma 1 demonstrates, in the first case where $t \notin \mathcal{B}$, we have $\eta_t = \sigma_t$. Moreover, since we set $\sigma_t = \eta_{t-1}/\beta$ for $t \ge 1$, in this case the step size will increase by a factor of $1/\beta$. In the second case

that $t \in \mathcal{B}$, our lower bound on the step size $\eta_t$ depends inversely on the normalized approximation error $\mathcal{E}_t = \frac{\|\mathbf{g}(\tilde{\mathbf{x}}_t) - \mathbf{g}(\mathbf{x}_t) - \hat{\mathbf{H}}_t(\tilde{\mathbf{x}}_t - \mathbf{x}_t)\|}{\|\tilde{\mathbf{x}}_t - \mathbf{x}_t\|}$. Also, note that $\mathcal{E}_t$ involves an auxiliary iterate $\tilde{\mathbf{x}}_t$ instead of the actual iterate $\hat{\mathbf{x}}_t$ accepted by our line search. We use the first result to relate $\|\tilde{\mathbf{x}}_t - \mathbf{x}_t\|$ to $\|\hat{\mathbf{x}} - \mathbf{x}_t\|$.

To shed light on our analysis, we use the triangle inequality and decompose this error into two terms:

$$\mathcal{E}_t \leq \frac{\|\mathbf{g}(\tilde{\mathbf{x}}_t) - \mathbf{g}(\mathbf{x}_t) - \mathbf{H}_t(\tilde{\mathbf{x}}_t - \mathbf{x}_t)\|}{\|\tilde{\mathbf{x}}_t - \mathbf{x}_t\|} + \|\mathbf{H}_t - \tilde{\mathbf{H}}_t\|. \tag{8}$$

The first term in (8) represents the intrinsic error from the linear approximation in the inexact proximal update, while the second term arises from the Hessian approximation error. Using the smoothness properties of $f$, we can upper bound the first term, as shown in the following lemma.

**Lemma 2.** *Under Assumptions 2 and 3, we have*

$$\frac{\|\mathbf{g}(\tilde{\mathbf{x}}_t) - \mathbf{g}(\mathbf{x}_t) - \mathbf{H}_t(\tilde{\mathbf{x}}_t - \mathbf{x}_t)\|}{\|\tilde{\mathbf{x}}_t - \mathbf{x}_t\|} \leq \min\left\{M_1, \frac{L_2\|\mathbf{x}_t - \mathbf{x}^*\|}{2\beta\sqrt{1-\alpha^2}}\right\}. \tag{9}$$

Lemma 2 shows that the linear approximation error is upper bounded by $M_1$. Moreover, the second upper bound is $O(\|\mathbf{x}_t - \mathbf{x}^*\|)$. Thus, as Algorithm 1 converges to the optimal solution $\mathbf{x}^*$, the second bound in (9) will become tighter than the first one, and the right hand side approaches zero.

To analyze the second term in (8), we isolate the noise component in our averaged Hessian estimate. Specifically, recall $\tilde{\mathbf{H}}_t = \sum_{i=0}^{t} z_{i,t}\hat{\mathbf{H}}_i$ and $\hat{\mathbf{H}}_i = \mathbf{H}_i + \mathbf{E}_i$. Thus, we have $\tilde{\mathbf{H}}_t = \bar{\mathbf{H}}_t + \bar{\mathbf{E}}_t$, where $\bar{\mathbf{H}}_t = \sum_{i=0}^{t} z_{i,t}\mathbf{H}_i$ is the aggregated Hessian and $\bar{\mathbf{E}}_t = \sum_{i=0}^{t} z_{i,t}\mathbf{E}_i$ is the aggregated Hessian noise, and it follows from the triangle inequality that $\|\mathbf{H}_t - \tilde{\mathbf{H}}_t\| \leq \|\mathbf{H}_t - \bar{\mathbf{H}}_t\| + \|\bar{\mathbf{E}}_t\|$. We refer to the first part, $\|\mathbf{H}_t - \bar{\mathbf{H}}_t\|$, as the *bias* of our Hessian estimate, and the second part, $\|\bar{\mathbf{E}}_t\|$, as the *averaged stochastic error*. There is an intrinsic trade-off between the two error terms. For the fastest error concentration, we assign equal weights to all past stochastic Hessian noises, i.e., $z_{i,t} = 1/(t+1)$ for all $0 \leq i \leq t$, corresponding to the uniform averaging scheme discussed in Section 4. To eliminate bias, we assign all weights to the most recent Hessian matrix $\mathbf{H}_t$, i.e., $z_{t,t} = 1$ and $z_{i,t} = 0$ for all $i < t$, but this incurs a large stochastic error. To balance these, we present a weighted averaging scheme in Section 5, gradually assigning more weight to recent stochastic Hessian approximations.

## 4 Analysis of uniform Hessian averaging

In this section, we present the convergence analysis of the uniform Hessian averaging scheme, where $w_t = t + 1$. In this case, we have $\tilde{\mathbf{H}}_t = \frac{1}{t+1}\sum_{i=0}^{t} \hat{\mathbf{H}}_i$. As discussed in Section 3.1, our main task is to lower bound the step size $\eta_t$, which requires us to control the approximation error $\mathcal{E}_t$ by analyzing the two error terms in (8). The first term is bounded by Lemma 2, and the second term can be bounded as $\|\mathbf{H}_t - \tilde{\mathbf{H}}_t\| \leq \|\mathbf{H}_t - \bar{\mathbf{H}}_t\| + \|\bar{\mathbf{E}}_t\|$. Next, we establish a bound on $\|\bar{\mathbf{E}}_t\|$, referred to as the Averaged Stochastic Error, and a bound on $\|\mathbf{H}_t - \bar{\mathbf{H}}_t\|$, referred to as the Bias Term.

**Averaged stochastic error.** To control the averaged Hessian noise $\|\bar{\mathbf{E}}_t\|$, we rely on the concentration of sub-exponential martingale difference, as shown in [1].

**Lemma 3** ([1, Lemma 2]). *Let $\delta \in (0,1)$ with $d/\delta \geq e$. Then with probability $1 - \delta\pi^2/6$, we have* $\|\bar{\mathbf{E}}_t\| \leq 8\Upsilon_E\sqrt{\frac{\log(d(t+1)/\delta)}{t+1}}$ *for any* $t \geq 4\log(d/\delta)$.

Lemma 3 shows that, with high probability, the norm of averaged Hessian noise $\|\bar{\mathbf{E}}_t\|$ approaches zero at the rate of $\tilde{\mathcal{O}}(\Upsilon_E/\sqrt{t})$. As discussed in Section 4.1, this error eventually becomes the dominant factor in the approximation error $\mathcal{E}_t$ and determines the final superlinear rate of our algorithm.

*Remark* 5. Our subsequent results are conditioned on the event that the bound on $\|\bar{\mathbf{E}}_t\|$ stated in Lemma 3 is satisfied for all $t \geq 4\log(d/\delta)$. Thus, to avoid redundancy, we will omit the "with high probability" qualification in the following discussion.

**Bias.** We proceed to establish an upper bound on $\|\mathbf{H}_t - \bar{\mathbf{H}}_t\|$. The proof can be found in Appendix B.1.

**Lemma 4.** *If $\tilde{\mathbf{H}}_t = \frac{1}{t+1}\sum_{i=0}^{t} \hat{\mathbf{H}}_i$, then $\|\mathbf{H}_t - \bar{\mathbf{H}}_t\| \leq \frac{1}{t+1}\sum_{i=0}^{t} \|\mathbf{H}_t - \mathbf{H}_i\|$. Moreover, for any $i \geq 0$, we have $\|\mathbf{H}_t - \mathbf{H}_i\| \leq \max\{M_1, 2L_2\|\mathbf{x}_i - \mathbf{x}^*\|\}$.*

The analysis of the bias term is more complicated. Specifically, to obtain the best result, we break the sum in Lemma 4 into two parts, $\frac{1}{t}\sum_{i=0}^{\mathcal{I}-1}\|\mathbf{H}_t - \mathbf{H}_i\|$ and $\frac{1}{t}\sum_{i=\mathcal{I}}^{t}\|\mathbf{H}_t - \mathbf{H}_i\|$, where $\mathcal{I}$ is an integer to be specified later. The first part corresponds to the bias from stale Hessian information and converges to zero at $\mathcal{O}(M_1\mathcal{I}/t)$, as shown by the first bound in Lemma 4. The second part is the bias from recent Hessian information when the iterates are near the optimal solution $\mathbf{x}^*$. Using the second bound in Lemma 4, we show this part contributes less to the total bias and is dominated by the first part. Thus, we can conclude that $\|\mathbf{H}_t - \bar{\mathbf{H}}_t\| = \mathcal{O}(\frac{M_1\mathcal{I}}{t+1})$.

Based on the previous discussions, it is evident that the terms contributing to the upper bound of $\mathcal{E}_t$ all converge to zero, albeit at different rates. Furthermore, the linear approximation error and bias term display distinct global and local convergence patterns, depending on the distance $\|\mathbf{x}_t - \mathbf{x}^*\|$. Hence, this necessitates a multi-phase convergence analysis, which we undertake in the following section.

## 4.1 Convergence analysis

Similar to [1], we consider four convergence phases with three transitions points $\mathcal{T}_1$, $\mathcal{T}_2$, and $\mathcal{T}_3$, whose expressions will be specified later. Due to space limitations, in the following we provide an overview of the four phases and relegate the details to Appendix B.

**Warm-up phase** $0 \leq t < \mathcal{T}_1$. At the beginning of the algorithm, the averaged Hessian estimate is dominated by stochastic noise and provides little useful information for convergence. Thus, there are generally no guarantees on the convergence rate for $0 \leq t < \mathcal{T}_1$. However, due to the line search scheme, Proposition 1 ensures that the distance to $\mathbf{x}^*$ is non-increasing, i.e., $\|\mathbf{x}_{t+1} - \mathbf{x}^*\| \leq \|\mathbf{x}_t - \mathbf{x}^*\|$ for all $t \geq 0$. During the warm-up phase, the averaged Hessian noise $\|\bar{\mathbf{E}}_t\|$, which contributes most to the approximation error $\mathcal{E}_t$, is gradually suppressed. Once the averaged Hessian noise is sufficiently concentrated, Algorithm 1 transitions to the linear convergence phase, denoted by $\mathcal{T}_1$.

**Linear convergence phase** $\mathcal{T}_1 \leq t < \mathcal{T}_2$. After $\mathcal{T}_1$ iterations, Algorithm 1 starts converging linearly to the optimal solution $\mathbf{x}^*$. Moreover, during this phase, all the three errors discussed in Section 3.1 continue to decrease. Specifically, Lemma 2 shows the linear approximation error is bounded by $\mathcal{O}(\|\mathbf{x}_t - \mathbf{x}^*\|)$, which converges to zero at a linear rate. Furthermore, Lemma 3 implies that the averaged Hessian error $\|\bar{\mathbf{E}}_t\|$ diminishes at a rate of $\tilde{\mathcal{O}}(\frac{\Upsilon_E}{\sqrt{t}})$. Finally, regarding the bias term, it can be shown $\|\mathbf{H}_t - \bar{\mathbf{H}}_t\| = \mathcal{O}(\frac{M_1\mathcal{I}}{t})$ following the discussions after Lemma 4. Thus, once all the three errors are sufficiently small, Algorithm 1 moves to the superlinear phase, denoted by $\mathcal{T}_2$.

**Superlinear phases** $\mathcal{T}_2 \leq t < \mathcal{T}_3$ **and** $\mathcal{T}_3 \leq t < \mathcal{T}_4$. After $\mathcal{T}_2$ iterations, Algorithm 1 converges at a superlinear rate. Moreover, the superlinear rate is determined by the averaged noise $\|\bar{\mathbf{E}}_t\|$, which decays at the rate of $\tilde{\mathcal{O}}(\frac{\Upsilon_E}{\sqrt{t}})$, and the bias of our averaged Hessian estimate $\tilde{\mathbf{H}}_t$, which decays at the rate of $\mathcal{O}(\frac{M_1\mathcal{I}}{t})$. Hence, as the number of iterations $t$ increases, the averaged noise will dominate and the algorithm transitions from the initial superlinear rate to the final superlinear rate.

We summarize our convergence guarantees in the following theorem and the proofs are in Appendix B.

**Theorem 1.** *Suppose Assumptions 1-5 hold and the weights for Hessian averaging in SNPE are uniform, and define $C := \frac{1}{2\beta\sqrt{1-\alpha^2}} + 5$. Then, the followings hold:*

(a) **Warm-up phase:** *If $0 \leq t < \mathcal{T}_1$, then $\|\mathbf{x}_{t+1} - \mathbf{x}^*\| \leq \|\mathbf{x}_t - \mathbf{x}^*\|$, where $\mathcal{T}_1 = \tilde{\mathcal{O}}(\frac{\Upsilon^2}{\kappa^2})$.*

(b) **Linear convergence phase:** *If $\mathcal{T}_1 \leq t < \mathcal{T}_2$, then $\|\mathbf{x}_{t+1} - \mathbf{x}^*\|^2 \leq \|\mathbf{x}_t - \mathbf{x}^*\|^2(1 + \frac{2\alpha\beta}{3\kappa})^{-1}$, where $\mathcal{T}_2 = \tilde{\mathcal{O}}(\max\{\frac{\Upsilon^2}{\kappa} + \kappa^2, \Upsilon^2\}) = \tilde{\mathcal{O}}(\Upsilon^2 + \kappa^2)$.*

(c) **Initial superlinear phase:** *For $\mathcal{T}_2 \leq t < \mathcal{T}_3$, we have $\|\mathbf{x}_{t+1} - \mathbf{x}^*\| \leq C\rho_t^{(1)}\|\mathbf{x}_t - \mathbf{x}^*\|$, where $\rho_t^{(1)} = \frac{6\kappa\mathcal{I}}{\alpha\sqrt{2\beta}(t+1)} = \tilde{\mathcal{O}}\left(\frac{\Upsilon^2/\kappa + \kappa^2}{t}\right)$ with $\mathcal{I}$ defined in (28) and $\mathcal{T}_3 = \tilde{\mathcal{O}}(\frac{(\Upsilon^2/\kappa + \kappa^2)^2}{\Upsilon^2})$.*

(d) **Final superlinear phase:** *Finally, for $t \geq \mathcal{T}_3$, we have $\|\mathbf{x}_{t+1} - \mathbf{x}^*\| \leq C\rho_t^{(2)}\|\mathbf{x}_t - \mathbf{x}^*\|$, where $\rho_t^{(2)} = \frac{8\sqrt{2}\Upsilon}{\alpha\sqrt{\beta}}\sqrt{\frac{\log(d(t+1)/\delta)}{t+1}} = \mathcal{O}\left(\Upsilon\sqrt{\frac{\log(t)}{t}}\right)$.*

**Comparison with [1]**. As shown in Table 1, our method in Algorithm 1 with uniform averaging achieves the same final superlinear convergence rate as the stochastic Newton method in [1]. However, it transitions to the linear and superlinear phases much earlier. Specifically, the initial transition point

$\mathcal{T}_1$ is improved by a factor of $\kappa^2$, and our linear rate in Lemma 6 is faster. This reduces the iterations needed to reach the local neighborhood, cutting the time to reach $\mathcal{T}_2$ and $\mathcal{T}_3$ by factors of $\kappa$ and $\kappa^2$.

## 5 Analysis of weighted Hessian averaging

Previously, we showed Algorithm 1 with uniform averaging eventually achieves superlinear convergence. However, as per Theorem 1, it requires $\tilde{\mathcal{O}}(\frac{\kappa^4}{\Upsilon^2})$ iterations to reach this rate. To achieve a faster transition, we follow [1] and use Hessian averaging with a general weight sequence $\{w_t\}$. We show this method also outperforms the stochastic Newton method in [1]. Specifically, we set $w_t = w(t)$ for all integer $t \geq 0$, where $w(\cdot) : \mathbb{R} \to \mathbb{R}$ satisfies certain regularity conditions as in [1, Assumption 3].

**Assumption 6.** *(i) $w(\cdot)$ is twice differentiable; (ii) $w(-1) = 0$, $w(t) > 0$, $\forall t \geq 0$; (iii) $w'(-1) \geq 0$; (iv) $w''(t) \geq 0$, $\forall t \geq -1$; (v) $\max\left\{\frac{w(t+1)}{w(t)}, \frac{w'(t+1)}{w'(t)}\right\} \leq \Psi$, $\forall t \geq 0$ for some $\Psi \geq 1$.*

Choosing $w(t) = t^p$ for any $p \geq 1$ satisfies Assumption 6. Additionally, as discussed in [1], a suitable choice is $w(t) = (t+1)^{\log(t+4)}$, allowing us to achieve the optimal transition to the superlinear rate. Since the analysis in this section closely resembles that in Section 4 on uniform averaging, we will only present the final result here for brevity. The four stages of convergence are detailed in the following theorem, with intermediate lemmas and proofs in the appendix. To simplify our bounds, we report results for non-uniform averaging with $w(t) = (t+1)^{\log(t+4)}$.

**Theorem 2.** *Suppose Assumptions 1-5 hold and the weights for Hessian averaging in SNPE are defined as $w(t) = (t+1)^{\log(t+4)}$, and define $C' := \left(\frac{1}{10\beta\sqrt{2(1-\alpha^2)}} + \frac{1}{\sqrt{2}}\right)$. Then, the following hold:*

(a) **Warm-up phase:** *If $0 \leq t < \mathcal{U}_1$, then $\|\mathbf{x}_{t+1} - \mathbf{x}^*\| \leq \|\mathbf{x}_t - \mathbf{x}^*\|$, where $\mathcal{U}_1 = \tilde{\mathcal{O}}(\frac{\Upsilon^2}{\kappa^2})$.*

(b) **Linear convergence phase:** *If $\mathcal{U}_1 \leq t < \mathcal{U}_2$, then $\|\mathbf{x}_{t+1} - \mathbf{x}^*\|^2 \leq \|\mathbf{x}_t - \mathbf{x}^*\|^2(1 + \frac{2\alpha\beta}{3\kappa})^{-1}$, where $\mathcal{U}_2 = \tilde{\mathcal{O}}(\max\{\frac{\Upsilon^2}{\kappa^2} + \kappa, \Upsilon^2\}) = \tilde{\mathcal{O}}(\Upsilon^2 + \kappa)$.*

(c) **Initial superlinear phase:** *If $\mathcal{U}_2 \leq t < \mathcal{U}_3$, then $\|\mathbf{x}_{t+1} - \mathbf{x}^*\| \leq C'\theta_t^{(1)}\|\mathbf{x}_t - \mathbf{x}^*\|$, where $\theta_t^{(1)} = \frac{5\kappa w(\mathcal{J})}{\alpha\sqrt{2\beta}w(t)} = \tilde{\mathcal{O}}\left(\kappa(\Upsilon^2+\kappa)^{\log(\Upsilon^2+\kappa)}/t^{\log t}\right)$ with $\mathcal{J}$ defined in (49) and $\mathcal{U}_3 = \tilde{O}(\Upsilon^2+\kappa)$.*

(d) **Final superlinear phase:** *Finally, if $t \geq \mathcal{U}_3$, then $\|\mathbf{x}_{t+1} - \mathbf{x}^*\| \leq C'\theta_t^{(2)}\|\mathbf{x}_t - \mathbf{x}^*\|$, where*
$$\theta_t^{(2)} = \frac{8\sqrt{2}\Upsilon}{\alpha\sqrt{\beta}}\sqrt{\frac{w'(t)\log(d\frac{t+1}{\delta})}{w(t)}} = \mathcal{O}\left(\frac{\Upsilon\log(t)}{\sqrt{t}}\right).$$

In the weighted averaging case, similar to the uniform averaging scenario, we observe four distinct phases of convergence. The warm-up phase for SNPE, during which the distance to the optimal solution does not increase, has the same duration as in the uniform averaging case but is shorter than the warm-up phase for the stochastic Newton method in [1] by a factor of $1/\kappa^2$. The linear convergence rates of both uniform and weighted Hessian averaging methods are $1 - \kappa^{-1}$, improving over the $1 - \kappa^{-2}$ rate achieved by the stochastic Newton method in [1]. The number of iterations to reach the initial superlinear phase is $\tilde{\mathcal{O}}(\Upsilon^2 + \kappa)$, smaller than the $\tilde{\mathcal{O}}(\kappa^2)$ needed for uniform averaging in SNPE when we focus on the regime where $\Upsilon = \mathcal{O}(\sqrt{\kappa})$. The non-uniform averaging method in [1] requires $\kappa^2$ iterations to achieve the initial superlinear phase, whereas the non-uniform SNPE achieves an initial superlinear rate of $\mathcal{O}(\kappa^{\log(\kappa)+1}/t)^t$, improving over the rate of $\mathcal{O}(\kappa^{4\log(\kappa)+1}/t^{\log(t)})^t$ in [1]. Finally, while the ultimate superlinear rates in all cases are comparable at approximately $\tilde{\mathcal{O}}((1/\sqrt{t})^t)$, the non-uniform version of SNPE requires $\tilde{\mathcal{O}}(\Upsilon^2 + \kappa)$ iterations to attain this fast rate, whereas [1]'s non-uniform version requires $\tilde{\mathcal{O}}(\kappa^2)$ iterations, which is less favorable.

*Remark 6.* As discussed in [1, Example 1], with a subsampling size of $s$, we have $\Upsilon = \tilde{\mathcal{O}}(\kappa/s)$ for subsampled Newton. This implies that when $s = \tilde{\Omega}(\sqrt{\kappa})$, we achieve $\Upsilon = \mathcal{O}(\sqrt{\kappa})$.

## 6 Discussion and complexity comparison

In this section, we compare the complexity of our method with accelerated gradient descent (AGD), damped Newton, and stochastic Newton [1] methods. The iteration complexities of these methods are summarized in Table 2, which we use to establish their overall complexities. Note that since both stochastic Newton and our proposed SNPE method achieve superlinear convergence, their

Table 2: Comparisons in terms of overall iteration complexity to find an $\epsilon$-accurate solution.

| Methods | AGD | Damped Newton | Stochastic Newton [1] | SNPE (**Ours**) |
|---|---|---|---|---|
| Iteration | $\mathcal{O}(\sqrt{\kappa}\log(1/\epsilon))$ | $\mathcal{O}(\kappa^2 + \log\log(1/\epsilon))$ | $\mathcal{O}(\Upsilon^2 + \kappa^2 + \frac{\log(1/\epsilon)}{\log\log(1/\epsilon)})$ | $\mathcal{O}(\Upsilon^2 + \kappa + \frac{\log(1/\epsilon)}{\log\log(1/\epsilon)})$ |

complexities depend on the target accuracy $\epsilon$ in the form $\mathcal{O}(\frac{\log(1/\epsilon)}{\log\log(1/\epsilon)})$, which is provably better than the complexity of AGD by at least a factor of $\log\log(\epsilon^{-1})$. Further details are provided in Appendix D.1. To better compare them, we focus on the finite-sum problem with $n$ functions: $\min_{x \in \mathbb{R}^d} \sum_{i=1}^n f_i(x)$. Let $\epsilon$ be the target accuracy, $\kappa$ the condition number, and $\Upsilon$ the noise level in Assumption 4. In this case, computing the exact gradient and Hessian costs $\mathcal{O}(nd)$ and $\mathcal{O}(nd^2)$, respectively. Thus, the per-iteration cost for AGD is $\mathcal{O}(nd)$. Each iteration of damped Newton's method requires computing the full Hessian and solving a linear system, resulting in a total per-iteration cost of $\mathcal{O}(nd^2 + d^3)$. For both stochastic Newton in [1] and our SNPE method, the per-iteration cost depends on how the stochastic Hessian is constructed. For example, Subsampled Newton constructs the Hessian estimate with a cost of $\mathcal{O}(sd^2)$, where $s$ denotes the sample size. Newton Sketch has a similar computation cost (see [6]). Additionally, it takes $\mathcal{O}(nd)$ to compute the full gradient and $\mathcal{O}(d^3)$ to solve the linear system. Since the sample/sketch size $s$ is typically chosen as $s = \mathcal{O}(d)$, the total per-iteration cost is $\mathcal{O}(nd + d^3)$.

- Compared to AGD, SNPE achieves better iteration complexity. Specifically, when the noise level $\Upsilon$ and target accuracy $\epsilon$ are relatively small ($\Upsilon = \mathcal{O}(\sqrt{\kappa})$ and $\log\frac{1}{\epsilon} = \Omega(\sqrt{\kappa})$), SNPE converges in fewer iterations. Additionally, when $n \geq d^2$ (indicating many samples), the per-iteration costs of both methods are $\mathcal{O}(nd)$, giving our method a better overall complexity.

- Compared to damped Newton's method, our method's iteration complexity depends better on the condition number $\kappa$, while damped Newton's depends better on $\epsilon$. However, when $n \geq d^2$, the per-iteration cost of damped Newton is $\mathcal{O}(nd^2)$, significantly more than our method's $\mathcal{O}(nd)$.

- Compared to the stochastic Newton method in [1], the per-iteration costs of both methods are similar. However, Table 2 shows that our iteration complexity is strictly better. Specifically, when the noise level is relatively small compared to the condition number, i.e., $\Upsilon = \mathcal{O}(\sqrt{\kappa})$, the complexity of SNPE improves by an additional factor of $\kappa$ over the stochastic Newton method.

## 7 Numerical experiments

While our focus is on improving theoretical guarantees, we also provide simple experiments to showcase our method's improvements. All simulations are implemented on a Windows PC with an AMD processor and 16GB Memory. We consider minimizing the regularized log-sum-exp objective $f(x) = \rho \log(\sum_{i=1}^n \exp(\frac{\mathbf{a}_i^\top \mathbf{x} - b_i}{\rho})) + \frac{\lambda}{2}\|\mathbf{x}\|^2$, a common test function for second-order methods [23, 24] due to its high ill-conditioning. Here, $\lambda > 0$ is a regularization parameter, $\rho > 0$ is a smoothing parameter, and the entries of the vectors $\mathbf{a}_1, \ldots, \mathbf{a}_n \in \mathbb{R}^d$ and $\mathbf{b} \in \mathbb{R}^d$ are randomly generated from the standard normal distribution and the uniform distribution over $[0, 1]$, respectively. In our experiments, the regularization parameter $\lambda$ is $10^{-3}$, the dimension $d$ is 500, and the number of samples $n$ is chosen from 50,000, 10,000, and 150,000, respectively.

We compare our SNPE method with the stochastic Newton method in [1], using both uniform Hessian averaging (Section 4) and weighted averaging (Section 5). In addition, we evaluate it against accelerated gradient descent (AGD), damped Newton's method, and Newton Proximal Extragradient (NPE), which corresponds to our SNPE method with the exact Hessian. For the stochastic Hessian estimate, we use a subsampling strategy with a subsampling size of $s = 500$. Empirically, we found that the extragradient step (Line 7 in Algorithm 1) tends to slow down convergence. To address this, we consider a variant of our method without the extragradient step, modifying Line 7 to $\mathbf{x}_{k+1} = \hat{\mathbf{x}}_k$. In Figure 3 of the Appendix, we further explore the effect of the extragradient step. We also note that similar observations were made in [19], where the simple iteration $\mathbf{x}_{t+1} = \mathbf{x}_t - \eta_t (\mathbf{I} + \eta_t \mathbf{H}_t)^{-1} \mathbf{g}_t$ outperformed "accelerated" second-order methods. From Figures 1 and 2, we have the following observations:

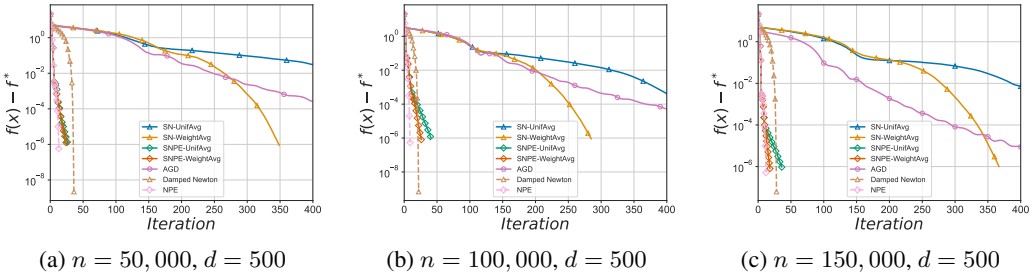

(a) $n = 50,000, d = 500$     (b) $n = 100,000, d = 500$     (c) $n = 150,000, d = 500$

Figure 1: Iteration complexity comparison for minimizing log-sum-exp on a synthetic dataset.

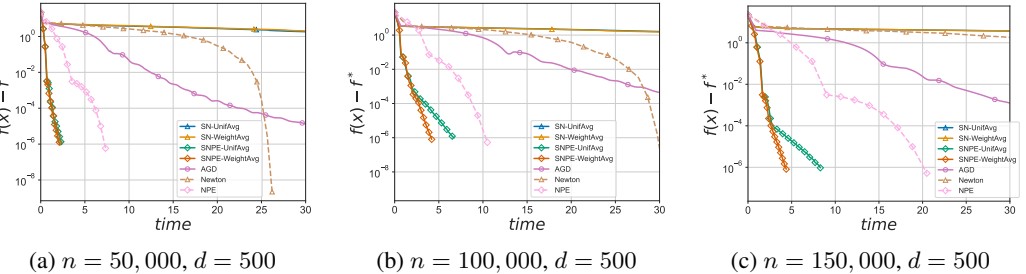

(a) $n = 50,000, d = 500$     (b) $n = 100,000, d = 500$     (c) $n = 150,000, d = 500$

Figure 2: Runtime comparison for minimizing log-sum-exp on a synthetic dataset.

**Comparison with stochastic Newton.** In all cases, our SNPE method outperforms stochastic Newton in both the number of iterations and runtime, due to the problem's highly ill-conditioned nature and our method's better dependence on the condition number.

**Comparison with AGD.** From Figure 1, we observe that our SNPE method, with either uniform or weighted averaging, requires far fewer iterations to converge than AGD due to the use of second-order information. Consequently, while SNPE has a higher per-iteration cost than AGD, it converges faster overall in terms of runtime, as demonstrated in Figure 2.

**Comparison with the damped Newton's method and NPE**. As expected, since both damped Newton and NPE use exact Hessian, Figure 1 shows that they exhibit superlinear convergence and converge in fewer iterations than the other algorithms. However, since the exact Hessian matrix is expensive to compute, they incur a high per-iteration computational cost and overall take more time than our proposed SNPE method to converge (see Figure 2). Moreover, the gap between these two methods and SNPE widens as the number of samples $n$ increases, demonstrating the advantage of our method in the large data regime.

## 8 Conclusions and limitations

We introduced a stochastic variant of the Newton Proximal Extragradient method (SNPE) for minimizing a strongly convex and smooth function with access to a noisy Hessian. Our contributions include establishing convergence guarantees under two Hessian averaging schemes: uniform and non-uniform. We characterized the computational complexity in both cases and demonstrated that SNPE outperforms the best-known results for the considered problem. A limitation of our theory is the assumption of strong convexity. Extending the theory to the convex setting would make it more general. This extension is left for future work due to space limitations.

## Acknowledgments

The research of R. Jiang and A. Mokhtari is supported in part by the NSF Grant 2007668, the NSF AI Institute for Foundations of Machine Learning (IFML), and the Wireless Networking and Communications Group (WNCG) Industrial Affiliates Program at UT Austin. The research of M. Dereziński was partially supported by NSF grant CCF-2338655.

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

# Appendix

## A Missing proofs in Section 3

### A.1 Proof of Proposition 1

In this section, we prove Proposition 1. In fact, using the same proof, we can also show an additional result that upper bounds $\|\mathbf{x}_t - \hat{\mathbf{x}}_t\|$, which will useful in the proof of Lemma 2. Therefore, we present the full version below for completeness.

**Proposition 2** (Full version of Proposition 1). *Let $\{\mathbf{x}_t\}_{t\geq 0}$ and $\{\hat{\mathbf{x}}_t\}_{t\geq 0}$ be the iterates generated by Algorithm 1. Then for any $t \geq 0$, we have $\|\mathbf{x}_{k+1} - \mathbf{x}^*\|^2 \leq \|\mathbf{x}_k - \mathbf{x}^*\|^2(1 + 2\eta_k\mu)^{-1}$ and $\|\mathbf{x}_t - \hat{\mathbf{x}}_t\| \leq \frac{1}{\sqrt{1-\alpha^2}}\|\mathbf{x}_t - \mathbf{x}^*\|$.*

*Proof.* Our proof is inspired by the approach in [22, Proposition 1]. For any $\mathbf{x} \in \mathbb{R}^d$, we first write

$$\eta_t\langle \nabla f(\hat{\mathbf{x}}_t), \hat{\mathbf{x}}_t - \mathbf{x}\rangle = \langle \hat{\mathbf{x}}_t - \mathbf{x}_t + \eta_t\nabla f(\hat{\mathbf{x}}_t), \hat{\mathbf{x}}_t - \mathbf{x}\rangle + \langle \mathbf{x}_t - \hat{\mathbf{x}}_t, \hat{\mathbf{x}}_t - \mathbf{x}\rangle. \tag{10}$$

To begin with, we bound the first term in (10) by

$$\begin{aligned}
\langle \hat{\mathbf{x}}_t - \mathbf{x}_t + \eta_t\nabla f(\hat{\mathbf{x}}_t), \hat{\mathbf{x}}_t - \mathbf{x}\rangle &\leq \|\hat{\mathbf{x}}_t - \mathbf{x}_t + \eta_t\nabla f(\hat{\mathbf{x}}_t)\|\|\hat{\mathbf{x}}_t - \mathbf{x}\| \\
&\leq \alpha\sqrt{1 + 2\eta_t\mu}\|\hat{\mathbf{x}}_t - \mathbf{x}_t\|\|\hat{\mathbf{x}}_t - \mathbf{x}\| \\
&\leq \frac{\alpha^2}{2}\|\hat{\mathbf{x}}_t - \mathbf{x}_t\|^2 + \frac{1 + 2\eta_t\mu}{2}\|\hat{\mathbf{x}}_t - \mathbf{x}\|^2,
\end{aligned} \tag{11}$$

where the first inequality is due to Cauchy-Schwarz inequality, the second inequality is due to the condition in (3), and the last inequality is due to Young's inequality. Moreover, for the second term in (10), we use the three-point equality to get

$$\langle \mathbf{x}_t - \hat{\mathbf{x}}_t, \hat{\mathbf{x}}_t - \mathbf{x}\rangle = \frac{1}{2}\|\mathbf{x}_t - \mathbf{x}\|^2 - \frac{1}{2}\|\mathbf{x}_t - \hat{\mathbf{x}}_t\|^2 - \frac{1}{2}\|\hat{\mathbf{x}}_t - \mathbf{x}\|^2. \tag{12}$$

By combining (10), (11) and (12), we obtain that

$$\eta_t\langle \nabla f(\hat{\mathbf{x}}_t), \hat{\mathbf{x}}_t - \mathbf{x}\rangle \leq \frac{1}{2}\|\mathbf{x}_t - \mathbf{x}\|^2 - \frac{1 - \alpha^2}{2}\|\mathbf{x}_t - \hat{\mathbf{x}}_t\|^2 + \eta_t\mu\|\hat{\mathbf{x}}_t - \mathbf{x}\|^2. \tag{13}$$

Moreover, it follows from the update rule in (4) that $\eta_t\nabla f(\hat{\mathbf{x}}_t) = \mathbf{x}_t - \mathbf{x}_{t+1} + 2\eta_t\mu(\hat{\mathbf{x}}_t - \mathbf{x}_{t+1})$. This implies that, for any $\mathbf{x} \in \mathbb{R}^d$,

$$\eta_t\langle \nabla f(\hat{\mathbf{x}}_t), \mathbf{x}_{t+1} - \mathbf{x}\rangle \tag{14}$$

$$= \langle \mathbf{x}_t - \mathbf{x}_{t+1}, \mathbf{x}_{t+1} - \mathbf{x}\rangle + 2\eta_t\mu\langle \hat{\mathbf{x}}_t - \mathbf{x}_{t+1}, \mathbf{x}_{t+1} - \mathbf{x}\rangle$$

$$= \frac{\|\mathbf{x}_t - \mathbf{x}\|^2}{2} - \frac{\|\mathbf{x}_t - \mathbf{x}_{t+1}\|^2}{2} - \frac{1 + 2\eta_t\mu}{2}\|\mathbf{x}_{t+1} - \mathbf{x}\|^2 + \eta_t\mu\|\hat{\mathbf{x}}_t - \mathbf{x}\|^2 - \eta_t\mu\|\hat{\mathbf{x}}_t - \mathbf{x}_{t+1}\|^2, \tag{15}$$

where we applied the three-point equality twice in the last equality. Thus, by combining (13) with $\mathbf{x} = \mathbf{x}_{t+1}$ and (15) with $\mathbf{x} = \mathbf{x}^*$, we get

$$\eta_t\langle \nabla f(\hat{\mathbf{x}}_t), \hat{\mathbf{x}}_t - \mathbf{x}^*\rangle$$

$$= \eta_t\langle \nabla f(\hat{\mathbf{x}}_t), \mathbf{x}_{t+1} - \mathbf{x}^*\rangle + \eta_t\langle \nabla f(\hat{\mathbf{x}}_t), \hat{\mathbf{x}}_t - \mathbf{x}_{t+1}\rangle$$

$$\leq \frac{\|\mathbf{x}_t - \mathbf{x}^*\|^2}{2} - \frac{\cancel{\|\mathbf{x}_t - \mathbf{x}_{t+1}\|^2}}{2} - \frac{1 + 2\eta_t\mu}{2}\|\mathbf{x}_{t+1} - \mathbf{x}^*\|^2 + \eta_t\mu\|\hat{\mathbf{x}}_t - \mathbf{x}^*\|^2 - \cancel{\eta_t\mu\|\hat{\mathbf{x}}_t - \mathbf{x}_{t+1}\|^2}$$

$$+ \frac{\cancel{\|\mathbf{x}_t - \mathbf{x}_{t+1}\|^2}}{2} - \frac{1 - \alpha^2}{2}\|\mathbf{x}_t - \hat{\mathbf{x}}_t\|^2 + \cancel{\eta_t\mu\|\hat{\mathbf{x}}_t - \mathbf{x}_{t+1}\|^2}. \tag{16}$$

Since $\nabla f(\mathbf{x}^*) = 0$ and $f$ is $\mu$-strongly convex, we further have

$$\langle \nabla f(\hat{\mathbf{x}}_t), \hat{\mathbf{x}}_t - \mathbf{x}^*\rangle = \langle \nabla f(\hat{\mathbf{x}}_t) - \nabla f(\mathbf{x}^*), \hat{\mathbf{x}}_t - \mathbf{x}^*\rangle \geq \mu\|\hat{\mathbf{x}}_t - \mathbf{x}^*\|^2. \tag{17}$$

Combining (16) and (17) and rearranging the terms, we obtain that

$$\frac{1 + 2\eta_t\mu}{2}\|\mathbf{x}_{t+1} - \mathbf{x}^*\|^2 \leq \frac{1}{2}\|\mathbf{x}_t - \mathbf{x}^*\|^2 - \frac{1 - \alpha^2}{2}\|\mathbf{x}_t - \hat{\mathbf{x}}_t\|^2. \tag{18}$$

Since $\alpha < 1$, the last term in (18) is negative and we immediately obtain that $\|\mathbf{x}_{t+1} - \mathbf{x}^*\|^2 \leq \|\mathbf{x}_t - \mathbf{x}^*\|^2(1 + 2\eta_t\mu)^{-1}$. Moreover, since $\frac{1 + 2\eta_t\mu}{2}\|\mathbf{x}_{t+1} - \mathbf{x}^*\|^2 \geq 0$, it follows that $\frac{1 - \alpha^2}{2}\|\mathbf{x}_t - \hat{\mathbf{x}}_t\|^2 \leq \frac{1}{2}\|\mathbf{x}_t - \mathbf{x}^*\|^2$, which leads to $\|\mathbf{x}_t - \hat{\mathbf{x}}_t\| \leq \frac{1}{\sqrt{1-\alpha^2}}\|\mathbf{x}_t - \mathbf{x}^*\|$. The proof is complete. □

## A.2 Proof of Lemma 1

Recall that in our backtracking line search scheme in Algorithm 1, the step size $\eta_t$ starts from $\sigma_t$ and keeps backtracking until the condition in (3) is satisfied. Hence, by the definition of $\mathcal{B}$, it immediately follows that $\eta_t = \sigma_t$ if $t \notin \mathcal{B}$. Moreover, if $t \in \mathcal{B}$, then the step size $\tilde{\eta}_t = \eta_t/\beta$ and the corresponding iterate $\tilde{\mathbf{x}}_t$ must have failed the condition in (3) (Otherwise, our line search scheme would have accepted the step size $\tilde{\eta}_t$ instead). This implies that

$$\|\tilde{\mathbf{x}}_t - \mathbf{x}_t + \tilde{\eta}_t \nabla f(\tilde{\mathbf{x}}_t)\| > \alpha\sqrt{1 + 2\tilde{\eta}_t\mu}\|\tilde{\mathbf{x}}_t - \mathbf{x}_t\|. \tag{19}$$

Since $\tilde{\mathbf{x}}_t = \mathbf{x}_t - \tilde{\eta}_t(\mathbf{I} + \tilde{\eta}_t\tilde{\mathbf{H}}_t)^{-1}\nabla f(\mathbf{x}_t)$, we have

$$\tilde{\mathbf{x}}_t - \mathbf{x}_t + \tilde{\eta}_t\tilde{\mathbf{H}}_t(\tilde{\mathbf{x}}_t - \mathbf{x}_t) = -\tilde{\eta}_t\nabla f(\mathbf{x}_t) \quad \Leftrightarrow \quad \tilde{\mathbf{x}}_t - \mathbf{x}_t = -\tilde{\eta}_t\big(\nabla f(\mathbf{x}_t) + \tilde{\mathbf{H}}_t(\tilde{\mathbf{x}}_t - \mathbf{x}_t)\big).$$

and hence the left-hand side in (19) equals to $\tilde{\eta}_t\|\nabla f(\tilde{\mathbf{x}}_t) - \nabla f(\mathbf{x}_t) - \tilde{\mathbf{H}}_t(\tilde{\mathbf{x}}_t - \mathbf{x}_t)\|$. Thus, we obtain from (19) that

$$\tilde{\eta}_t > \frac{\alpha\sqrt{1 + 2\tilde{\eta}_t\mu}\|\tilde{\mathbf{x}}_t - \mathbf{x}_t\|}{\|\nabla f(\tilde{\mathbf{x}}_t) - \nabla f(\mathbf{x}_t) - \tilde{\mathbf{H}}_t(\tilde{\mathbf{x}}_t - \mathbf{x}_t)\|}, \tag{20}$$

By substituting $\eta_t = \beta\tilde{\eta}_t$, we further have

$$\eta_t > \frac{\alpha\beta\sqrt{1 + 2\eta_t\mu/\beta}\|\tilde{\mathbf{x}}_t - \mathbf{x}_t\|}{\|\nabla f(\tilde{\mathbf{x}}_t) - \nabla f(\mathbf{x}_t) - \tilde{\mathbf{H}}_t(\tilde{\mathbf{x}}_t - \mathbf{x}_t)\|}.$$

Using the fact that $1 + 2\eta_t\mu/\beta \geq 1$, we get

$$\eta_t > \frac{\alpha\beta\|\tilde{\mathbf{x}}_t - \mathbf{x}_t\|}{\|\nabla f(\tilde{\mathbf{x}}_t) - \nabla f(\mathbf{x}_t) - \tilde{\mathbf{H}}_t(\tilde{\mathbf{x}}_t - \mathbf{x}_t)\|}. \tag{21}$$

Moreover, using the fact that $1 + 2\eta_t\mu/\beta \geq 2\eta_t\mu/\beta$, we can also conclude that

$$\eta_t > \frac{\alpha\beta\sqrt{2\eta_t\mu/\beta}\|\tilde{\mathbf{x}}_t - \mathbf{x}_t\|}{\|\nabla f(\tilde{\mathbf{x}}_t) - \nabla f(\mathbf{x}_t) - \tilde{\mathbf{H}}_t(\tilde{\mathbf{x}}_t - \mathbf{x}_t)\|} \quad \Rightarrow \quad \eta_t > \frac{2\alpha^2\beta\mu\|\tilde{\mathbf{x}}_t - \mathbf{x}_t\|^2}{\|\nabla f(\tilde{\mathbf{x}}_t) - \nabla f(\mathbf{x}_t) - \tilde{\mathbf{H}}_t(\tilde{\mathbf{x}}_t - \mathbf{x}_t)\|^2}. \tag{22}$$

By combining (21) and (22), we obtain the lower bound in Lemma 1.

Finally, when $\tilde{\mathbf{H}}_t \succeq 0$, it holds that $\mathbf{I} + \tilde{\eta}_t\tilde{\mathbf{H}}_t \succeq \mathbf{I} + \eta_t\tilde{\mathbf{H}}_t \succeq 0$ and thus $(\mathbf{I} + \eta_t\tilde{\mathbf{H}}_t)^{-1} \succeq (\mathbf{I} + \tilde{\eta}_t\tilde{\mathbf{H}}_t)^{-1} \succeq 0$. This further implies that $\|(\mathbf{I} + \eta_t\tilde{\mathbf{H}}_t)^{-1}\nabla f(\mathbf{x}_t)\| \geq \|(\mathbf{I} + \tilde{\eta}_t\tilde{\mathbf{H}}_t)^{-1}\nabla f(\mathbf{x}_t)\|$. Hence, we can conclude that

$$\|\tilde{\mathbf{x}}_t - \mathbf{x}_t\| = \tilde{\eta}_t\|(\mathbf{I} + \tilde{\eta}_t\tilde{\mathbf{H}}_t)^{-1}\nabla f(\mathbf{x}_t)\| \leq \frac{\eta_t}{\beta}\|(\mathbf{I} + \eta_t\tilde{\mathbf{H}}_t)^{-1}\nabla f(\mathbf{x}_t)\| \leq \frac{1}{\beta}\|\hat{\mathbf{x}}_t - \mathbf{x}_t\|.$$

This completes the proof.

## A.3 Extension to inexact linear solving

In this section, we extend our convergence results to the case where the linear system in (6) is solved inexactly, i.e., we find $\hat{\mathbf{x}}_t$ such that

$$\|(\mathbf{I} + \eta_t\tilde{\mathbf{H}}_t)(\hat{\mathbf{x}}_t - \mathbf{x}_t) + \eta_t\nabla f(\mathbf{x}_t)\| \leq \frac{\alpha}{2}\|\hat{\mathbf{x}}_t - \mathbf{x}_t\|. \tag{23}$$

In this case, since the proof of Proposition 2 does not rely on the update rule in (6), Proposition 2 continues to hold. However, we need to modify the proof of Lemma 1 and replace it by the following lemma. We note that the two results differ only by an absolute constant.

**Lemma 5** (Extension to Lemma 1). *For $t \notin \mathcal{B}$, we have $\eta_t = \sigma_t$. For $t \in \mathcal{B}$, let $\tilde{\eta}_t = \eta_t/\beta$ and $\tilde{\mathbf{x}}_t$ be the corresponding iterate rejected by our line search scheme. Then, $\|\tilde{\mathbf{x}}_t - \mathbf{x}_t\| \leq \frac{3}{\beta}\|\hat{\mathbf{x}}_t - \mathbf{x}_t\|$. Moreover,*

$$\eta_t \geq \max\left\{\frac{\alpha\beta\|\tilde{\mathbf{x}}_t - \mathbf{x}_t\|}{2\|\mathbf{g}(\tilde{\mathbf{x}}_t) - \mathbf{g}(\mathbf{x}_t) - \tilde{\mathbf{H}}_t(\tilde{\mathbf{x}}_t - \mathbf{x}_t)\|}, \frac{\alpha^2\beta\mu\|\tilde{\mathbf{x}}_t - \mathbf{x}_t\|^2}{2\|\mathbf{g}(\tilde{\mathbf{x}}_t) - \mathbf{g}(\mathbf{x}_t) - \tilde{\mathbf{H}}_t(\tilde{\mathbf{x}}_t - \mathbf{x}_t)\|^2}\right\}.$$

*Proof.* We follow a similar argument as in the proof of Lemma 1. If $t \notin \mathcal{B}$, it immediately follows that $\eta_t = \sigma_t$. Further, if $t \in \mathcal{B}$, then $\tilde{\eta}_t$ and the corresponding iterate $\tilde{x}_t$ must fail to satisfy the condition in (3). This implies that

$$\|\tilde{\mathbf{x}}_t - \mathbf{x}_t + \tilde{\eta}_t \nabla f(\tilde{\mathbf{x}}_t)\| > \alpha \sqrt{1 + 2\tilde{\eta}_t \mu} \|\tilde{\mathbf{x}}_t - \mathbf{x}_t\|. \tag{24}$$

Moreover, by our inexactness condition in (23), $\tilde{x}_t$ satisfies

$$\|(\mathbf{I} + \tilde{\eta}_t \tilde{\mathbf{H}}_t)(\tilde{\mathbf{x}}_t - \mathbf{x}_t) + \tilde{\eta}_t \nabla f(\mathbf{x}_t)\| \leq \frac{\alpha}{2} \|\tilde{\mathbf{x}}_t - \mathbf{x}_t\|.$$

Hence, by using triangle inequality, we have

$$\begin{aligned}
&\tilde{\eta}_t \|\nabla f(\tilde{\mathbf{x}}_t) - \nabla f(\mathbf{x}_t) - \tilde{\mathbf{H}}_t(\tilde{\mathbf{x}}_t - \mathbf{x}_t)\| \\
&= \left\| (\tilde{\mathbf{x}}_t - \mathbf{x}_t + \tilde{\eta}_t \nabla f(\tilde{\mathbf{x}}_t)) - \left( (\mathbf{I} + \tilde{\eta}_t \tilde{\mathbf{H}}_t)(\tilde{\mathbf{x}}_t - \mathbf{x}_t) + \tilde{\eta}_t \nabla f(\mathbf{x}_t) \right) \right\| \\
&\geq \|\tilde{\mathbf{x}}_t - \mathbf{x}_t + \tilde{\eta}_t \nabla f(\tilde{\mathbf{x}}_t)\| - \|(\mathbf{I} + \tilde{\eta}_t \tilde{\mathbf{H}}_t)(\tilde{\mathbf{x}}_t - \mathbf{x}_t) + \tilde{\eta}_t \nabla f(\mathbf{x}_t)\| \\
&\geq \frac{\alpha}{2} \sqrt{1 + 2\tilde{\eta}_t \mu} \|\tilde{\mathbf{x}}_t - \mathbf{x}_t\|.
\end{aligned}$$

Thus, we obtain that

$$\tilde{\eta}_t > \frac{\alpha \sqrt{1 + 2\tilde{\eta}_t \mu} \|\tilde{\mathbf{x}}_t - \mathbf{x}_t\|}{2\|\nabla f(\tilde{\mathbf{x}}_t) - \nabla f(\mathbf{x}_t) - \tilde{\mathbf{H}}_t(\tilde{\mathbf{x}}_t - \mathbf{x}_t)\|},$$

Combining this with (20), we observe that these two bounds differ only by a constant factor of 2. Hence, the rest of the proof for the lower bound follows similarly as in Lemma 1.

Next, we prove the inequality $\|\tilde{\mathbf{x}}_t - \mathbf{x}_t\| \leq \frac{1}{\beta} \|\hat{\mathbf{x}}_t - \mathbf{x}_t\|$. Define $\hat{\mathbf{x}}_t^* = \mathbf{x}_t - \eta_t(\mathbf{I} + \eta_t \tilde{\mathbf{H}}_t)^{-1} \nabla f(\mathbf{x}_t)$ and $\tilde{\mathbf{x}}_t^* = \mathbf{x}_t - \tilde{\eta}_t(\mathbf{I} + \tilde{\eta}_t \tilde{\mathbf{H}}_t)^{-1} \nabla f(\mathbf{x}_t)$, i.e, they are the exact solutions to the corresponding linear systems. By the argument in the proof of Lemma 1, we have $\|\tilde{\mathbf{x}}_t^* - \mathbf{x}_t\| \leq \frac{1}{\beta} \|\hat{\mathbf{x}}_t^* - \mathbf{x}_t\|$. In the following, we first prove that

$$\frac{1}{2}\|\hat{\mathbf{x}}_t - \mathbf{x}_t\| \leq \|\hat{\mathbf{x}}_t^* - \mathbf{x}_t\| \leq \frac{3}{2}\|\hat{\mathbf{x}}_t - \mathbf{x}_t\| \quad \text{and} \quad \frac{1}{2}\|\tilde{\mathbf{x}}_t - \mathbf{x}_t\| \leq \|\tilde{\mathbf{x}}_t^* - \mathbf{x}_t\| \leq \frac{3}{2}\|\tilde{\mathbf{x}}_t - \mathbf{x}_t\|. \tag{25}$$

It suffices to prove the first set of inequalities, since the second one follows similarly. To see this, note that the condition in (23) can be rewritten as

$$\|(\mathbf{I} + \eta_t \tilde{\mathbf{H}}_t)(\hat{\mathbf{x}}_t - \hat{\mathbf{x}}_t^*)\| \leq \frac{\alpha}{2} \|\hat{\mathbf{x}}_t - \mathbf{x}_t\|.$$

Since $\tilde{\mathbf{H}}_t \succeq 0$, this further implies that $\|\hat{\mathbf{x}}_t - \hat{\mathbf{x}}_t^*\| \leq \|(\mathbf{I} + \eta_t \tilde{\mathbf{H}}_t)(\hat{\mathbf{x}}_t - \hat{\mathbf{x}}_t^*)\| \leq \frac{\alpha}{2} \|\hat{\mathbf{x}}_t - \mathbf{x}_t\|$. Hence, by the triangle inequality, we obtain that

$$\|\hat{\mathbf{x}}_t^* - \mathbf{x}_t\| \leq \|\hat{\mathbf{x}}_t^* - \hat{\mathbf{x}}_t\| + \|\hat{\mathbf{x}}_t - \mathbf{x}_t\| \leq \left(1 + \frac{\alpha}{2}\right) \|\hat{\mathbf{x}}_t - \mathbf{x}_t\| \leq \frac{3}{2}\|\hat{\mathbf{x}}_t - \mathbf{x}_t\|,$$

$$\|\hat{\mathbf{x}}_t^* - \mathbf{x}_t\| \geq \|\hat{\mathbf{x}}_t^* - \hat{\mathbf{x}}_t\| - \|\hat{\mathbf{x}}_t - \mathbf{x}_t\| \geq \left(1 - \frac{\alpha}{2}\right) \|\hat{\mathbf{x}}_t - \mathbf{x}_t\| \geq \frac{1}{2}\|\hat{\mathbf{x}}_t - \mathbf{x}_t\|,$$

which lead to (25). Finally, we conclude that $\|\tilde{\mathbf{x}}_t - \mathbf{x}_t\| \leq 2\|\tilde{\mathbf{x}}_t^* - \mathbf{x}_t\| \leq \frac{2}{\beta}\|\hat{\mathbf{x}}_t^* - \mathbf{x}_t\| \leq \frac{3}{\beta}\|\hat{\mathbf{x}}_t - \mathbf{x}_t\|$. This completes the proof. $\qquad\square$

### A.4 The total complexity of line search

Let $l_t$ denote the number of line search steps in iteration $t$. We first note that $\eta_t = \sigma_t \beta^{l_t - 1}$ by our line search subroutine, which implies $l_t = \log_{1/\beta}(\sigma_t/\eta_t) + 1$. Moreover, recall that $\sigma_t = \eta_{t-1}/\beta$ for $t \geq 1$. Hence, the total number of line search steps after $t$ iterations can be bounded by (cf. [22, Lemma 22]):

$$\sum_{i=0}^{t-1} l_i = \sum_{i=0}^{t-1} \left[ \log_{1/\beta} \left( \frac{\sigma_i}{\eta_i} \right) + 1 \right] = 2t - 1 + \log \left( \frac{\sigma_0}{\eta_{t-1}} \right).$$

Moreover, it can be shown that $\eta_{t-1} \geq \alpha\beta/(3M_1)$ when $t = \tilde{\Omega}(\Upsilon^2/\kappa^2)$ in both the uniform averaging and the non-uniform averaging cases (cf. Corollaries 2 and 3). This implies that the total number of line search steps can be bounded by $2t - 1 + \log(3M_1\sigma_0/\alpha\beta)$.

## A.5 Proof of Lemma 2

Recall that we use $\mathbf{g}(\mathbf{x})$ and $\mathbf{H}(\mathbf{x})$ to denote the gradient $\nabla f(\mathbf{x})$ and the Hessian $\nabla^2 f(\mathbf{x})$, respectively. We first consider the inequality in (9). By the fundamental theorem of calculus, we can write

$$\nabla f(\tilde{\mathbf{x}}_t) - \nabla f(\mathbf{x}_t) = \int_0^1 \nabla^2 f(\mathbf{x}_t + \tau(\tilde{\mathbf{x}}_t - \mathbf{x}_t))(\tilde{\mathbf{x}}_t - \mathbf{x}_t) \, d\tau.$$

Therefore, we can further use the triangle inequality to get

$$\begin{aligned}
&\|\nabla f(\tilde{\mathbf{x}}_t) - \nabla f(\mathbf{x}_t) - \nabla^2 f(\mathbf{x}_t)(\tilde{\mathbf{x}}_t - \mathbf{x}_t)\| \\
&= \left\| \int_0^1 \left( \nabla^2 f(\mathbf{x}_t + \tau(\tilde{\mathbf{x}}_t - \mathbf{x}_t)) - \nabla^2 f(\mathbf{x}_t) \right)(\tilde{\mathbf{x}}_t - \mathbf{x}_t) \, d\tau \right\| \\
&\leq \int_0^1 \left\| \left( \nabla^2 f(\mathbf{x}_t + \tau(\tilde{\mathbf{x}}_t - \mathbf{x}_t)) - \nabla^2 f(\mathbf{x}_t) \right)(\tilde{\mathbf{x}}_t - \mathbf{x}_t) \right\| \, d\tau \\
&\leq \int_0^1 \left\| \nabla^2 f(\mathbf{x}_t + \tau(\tilde{\mathbf{x}}_t - \mathbf{x}_t)) - \nabla^2 f(\mathbf{x}_t) \right\| \|\tilde{\mathbf{x}}_t - \mathbf{x}_t\| \, d\tau.
\end{aligned} \tag{26}$$

Moreover, we have $\left\| \nabla^2 f(\mathbf{x}_t + \tau(\tilde{\mathbf{x}}_t - \mathbf{x}_t)) - \nabla^2 f(\mathbf{x}_t) \right\| \leq M_1$ for any $\tau \in [0, 1]$ by Assumption 2. Together with (26), this further implies that $\frac{\|\nabla f(\tilde{\mathbf{x}}_t) - \nabla f(\mathbf{x}_t) - \nabla^2 f(\mathbf{x}_t)(\tilde{\mathbf{x}}_t - \mathbf{x}_t)\|}{\|\tilde{\mathbf{x}}_t - \mathbf{x}_t\|} \leq M_1$, which proves the first bound in (9).

Next, we consider the second bound in (9). By Assumption 3, it follows from standard arguments that $\|\nabla f(\tilde{\mathbf{x}}_t) - \nabla f(\mathbf{x}_t) - \nabla^2 f(\mathbf{x}_t)(\tilde{\mathbf{x}}_t - \mathbf{x}_t)\| \leq \frac{L_2}{2} \|\tilde{\mathbf{x}}_t - \mathbf{x}_t\|^2$ (e.g., see [25, Lemma 1.2.4]). This leads to

$$\frac{\|\nabla f(\tilde{\mathbf{x}}_t) - \nabla f(\mathbf{x}_t) - \nabla^2 f(\mathbf{x}_t)(\tilde{\mathbf{x}}_t - \mathbf{x}_t)\|}{\|\tilde{\mathbf{x}}_t - \mathbf{x}_t\|} \leq \frac{L_2 \|\tilde{\mathbf{x}}_t - \mathbf{x}_t\|}{2} \leq \frac{L_2 \|\hat{\mathbf{x}}_t - \mathbf{x}_t\|}{2\beta} \leq \frac{L_2 \|\mathbf{x}_t - \mathbf{x}^*\|}{2\beta\sqrt{1-\alpha^2}},$$

where we used $\|\tilde{\mathbf{x}}_t - \mathbf{x}_t\| \leq \frac{1}{\beta} \|\hat{\mathbf{x}}_t - \mathbf{x}_t\|$ from Lemma 1 and $\|\hat{\mathbf{x}}_t - \mathbf{x}_t\| \leq \frac{1}{\sqrt{1-\alpha^2}} \|\mathbf{x}_t - \mathbf{x}^*\|$ from Proposition 2. This completes the proof.

# B  Missing Proofs in Section 4

## B.1  Proof of Lemma 4

Recall that in the case of uniform averaging, we have $\bar{\mathbf{H}}_t = \frac{1}{t+1} \sum_{i=0}^t \mathbf{H}_i$. Hence, it follows from Jensen's inequality that $\|\bar{\mathbf{H}}_t - \mathbf{H}_t\| \leq \frac{1}{t+1} \sum_{i=0}^t \|\mathbf{H}_i - \mathbf{H}_t\|$. To prove the second claim, note that Assumption 2 directly implies $\|\mathbf{H}_i - \mathbf{H}_t\| \leq M_1$. Moreover, we can use Assumption 3 and the triangle inequality to bound $\|\mathbf{H}_t - \mathbf{H}_i\| \leq L_2 \|\mathbf{x}_t - \mathbf{x}_i\| \leq L_2(\|\mathbf{x}_t - \mathbf{x}^*\| + \|\mathbf{x}_i - \mathbf{x}^*\|)$. Since $i \leq t$ and $\|\mathbf{x}_t - \mathbf{x}^*\|$ is non-increasing in $t$ by Proposition 1, we further have $\|\mathbf{x}_t - \mathbf{x}^*\| \leq \|\mathbf{x}_i - \mathbf{x}^*\|$, which proves $\|\mathbf{H}_t - \mathbf{H}_i\| \leq 2L_2 \|\mathbf{x}_i - \mathbf{x}^*\|$.

## B.2  Proof of Theorem 1

**Warm-up phase.** To determine the transition point $\mathcal{T}_1$, recall that both the linear approximation error and the bias term can be bounded by $M_1$ according to Lemmas 2 and 4. Since Lemma 3 shows that $\|\bar{\mathbf{E}}_t\| = \tilde{\mathcal{O}}(\Upsilon_E/\sqrt{t})$, we will have $\|\bar{\mathbf{E}}_t\| \leq M_1$ when $t = \tilde{\Omega}(\Upsilon_E^2/M_1^2) = \tilde{\Omega}(\Upsilon^2/\kappa^2)$. More specifically, the transition point is given by

$$\mathcal{T}_1 = \max\left\{ \frac{256\Upsilon^2}{\kappa^2} \log \frac{8d\Upsilon}{\kappa\delta}, 4\log \frac{d}{\delta}, \log_{\frac{1}{\beta}} \frac{\alpha\beta}{3M_1\sigma_0} \right\}, \tag{27}$$

where $\delta \in (0, 1)$ satisfies $d/\delta \geq e$, $\alpha, \beta \in (0, 1)$ are line-search parameters, and $\sigma_0$ is the initial step size.

**Linear convergence phase**. In the following lemma, we prove the linear convergence of Algorithm 1 with uniform averaging.

**Lemma 6.** *Assume that $\beta \leq 1/2$ and recall the definition of $\mathcal{T}_1$ in (27). For any $t \geq \mathcal{T}_1$, we have*

$$\|\mathbf{x}_{t+1} - \mathbf{x}^*\|^2 \leq \|\mathbf{x}_t - \mathbf{x}^*\|^2 \left(1 + \frac{2\alpha\beta}{3\kappa}\right)^{-1},$$

*where $\kappa \triangleq M_1/\mu$ is the condition number.*

*Proof.* See Appendix B.3. □

Now we discuss the transition point $\mathcal{T}_2$. At a high level, the algorithm transitions to the superlinear phase if all three errors discussed in Section 3.1 are reduced from $\mathcal{O}(M)$ to $\mathcal{O}(\mu)$. For this to happen, first the iterate $\mathbf{x}_t$ needs to reach a local neighborhood satisfying $\|\mathbf{x}_t - \mathbf{x}^*\| = \mathcal{O}(\mu/L_2)$. As a corollary of Lemma 6, this holds at most after an additional $\tilde{\mathcal{O}}(\kappa)$ iterations. Specifically, let $\nu \in (0,1)$ be a parameter and define

$$\mathcal{I} = \mathcal{T}_1 + 2\left(1 + \frac{3\kappa}{2\alpha\beta}\right)\log\frac{L_2 D}{\nu\mu}, \tag{28}$$

where $D = \|\mathbf{x}_0 - \mathbf{x}^*\|$ is the initial distance to the optimal solution. Then Lemma 6 implies that we have $\|\mathbf{x}_t - \mathbf{x}^*\| \leq \nu\mu/L_2$ for all $t \geq \mathcal{I}$. Moreover, Lemma 3 implies that the averaged Hessian noise satisfies $\|\bar{\mathbf{E}}_t\| = \mathcal{O}(\mu)$ when $t = \tilde{\Omega}(\Upsilon_E^2/\mu^2) = \tilde{\Omega}(\Upsilon^2)$. Finally, regarding the bias term, following the discussions after Lemma 4, it can be shown that $\|\mathbf{H}_t - \bar{\mathbf{H}}_t\| = \mathcal{O}\left(\frac{M_1 \mathcal{I}}{t+1}\right)$. Thus, $\|\mathbf{H}_t - \bar{\mathbf{H}}_t\| = \mathcal{O}(\mu)$ when $t = \Omega(\kappa\mathcal{I})$. Combining all pieces together, we formally define the second transition point by

$$\mathcal{T}_2 = \max\left\{\frac{256\Upsilon^2}{\nu^2}\log\frac{8d\Upsilon}{\delta\nu}, \frac{\kappa\mathcal{I}}{\nu} - 1\right\}. \tag{29}$$

Since $\mathcal{I} = \tilde{\mathcal{O}}(\mathcal{T}_1 + \kappa) = \tilde{\mathcal{O}}(\Upsilon^2/\kappa^2 + \kappa)$, we note that $\mathcal{T}_2 = \tilde{\mathcal{O}}(\max\{\Upsilon^2, \Upsilon^2/\kappa + \kappa^2\}) = \tilde{\mathcal{O}}(\Upsilon^2 + \kappa^2)$.

**Superlinear phase.** In the following theorem, we show that after $\mathcal{T}_2$ iterations, Algorithm 1 with uniform averaging converges at a superlinear rate. See Appendix B.4 for proof.

**Theorem 3.** *Let $\nu \in (0,1)$ be a parameter satisfying $\left(\frac{5}{2\alpha\beta\sqrt{(1-\alpha^2)\beta}} + \frac{25}{\alpha\sqrt{2\beta}}\right)\nu \leq 1$. and recall the definition of $\mathcal{T}_2$ in (29). Then for any $t \geq \mathcal{T}_2$,*

$$\|\mathbf{x}_{t+1} - \mathbf{x}^*\| \leq \left(\frac{1}{2\beta\sqrt{1-\alpha^2}} + 5\right)\rho_t\|\mathbf{x}_t - \mathbf{x}^*\|,$$

*where*

$$\rho_t = \frac{4\Upsilon}{\alpha\sqrt{\beta}}\sqrt{\frac{\log(d(t+1)/\delta)}{t+1}} + \frac{3\kappa\mathcal{I}}{2\alpha\sqrt{\beta}(t+1)}. \tag{30}$$

In Theorem 3, we observe that the rate $\rho_t$ in (30) goes to zero as the number of iterations $t$ increases, and thus it implies that the iterates converge to $\mathbf{x}^*$ superlinearly. Moreover, the rate $\rho_t$ consists of two terms. The first term comes from the averaged noise $\|\bar{\mathbf{E}}_t\|$, which decays at the rate of $\tilde{\mathcal{O}}(\Upsilon/\sqrt{t})$. In addition, the second term is due to the bias of our averaged Hessian estimate $\tilde{\mathbf{H}}_t$, which decays at the rate of $\mathcal{O}(\kappa\mathcal{I}/t)$. Hence, when $t$ is sufficiently large, the averaged noise will dominate and the superlinear rate settles for the slower rate of $\tilde{\mathcal{O}}(\Upsilon/\sqrt{t})$. Specifically, the algorithm transitions from the initial superlinear rate to the final superlinear rate when the two terms in (30) are balanced. Hence, we define the third transition point $\mathcal{T}_3$ as the root of

$$64(\mathcal{T}_3 + 1)\log(d(\mathcal{T}_3 + 1)/\delta) = \frac{9\kappa^2\mathcal{I}^2}{\Upsilon^2}.$$

Since $\mathcal{I} = \tilde{\mathcal{O}}(\Upsilon^2/\kappa^2 + \kappa)$, $\mathcal{T}_3 = \tilde{\mathcal{O}}((\Upsilon^2/\kappa + \kappa^2)^2/\Upsilon^2)$. We summarize our discussions in the following corollary.

**Corollary 1.** *For $\mathcal{T}_2 \leq t \leq \mathcal{T}_3 - 1$, we have*

$$\|\mathbf{x}_{t+1} - \mathbf{x}^*\| \leq \left(\frac{1}{2\beta\sqrt{1-\alpha^2}} + 5\right)\rho_t^{(1)}\|\mathbf{x}_t - \mathbf{x}^*\|,$$

where $\rho_t^{(1)} = \frac{6\kappa\mathcal{I}}{\alpha\sqrt{2\beta}(t+1)}$. Moreover, for $t \geq \mathcal{T}_3$, we have

$$\|\mathbf{x}_{t+1} - \mathbf{x}^*\| \leq \left(\frac{1}{2\beta\sqrt{1-\alpha^2}} + 5\right)\rho_t^{(2)}\|\mathbf{x}_t - \mathbf{x}^*\|,$$

where $\rho_t^{(2)} = \frac{8\sqrt{2}\Upsilon}{\alpha\sqrt{\beta}}\sqrt{\frac{\log(d(t+1)/\delta)}{t+1}}$.

### B.3 Proof of Lemma 6

We divide the proof of Lemma 6 into the following three steps. First, in Lemma 7, we provide a lower bound on the the step size $\eta_t$ in those iterations where our line search scheme backtracks the step size, i.e., $t \in \mathcal{B}$. Building on Lemma 7, we use induction in Lemma 8 to prove a lower bound for all $t \geq 0$. This allows us to establish $\eta_t = \Omega(1/M_1)$ for all $t \geq \mathcal{T}_1$ in Corollary 2, from which Lemma 6 immediately follows.

To simplify the notation, we define the function

$$\phi(t) = 8\Upsilon_E\sqrt{\frac{\log(d(t+1)/\delta)}{t+1}}. \tag{31}$$

Then Lemma 3 can be equivalently written as $\|\bar{\mathbf{E}}_t\| \leq \phi(t)$ for all $t \geq 4\log(d/\delta)$. We are now ready to state our first lemma.

**Lemma 7.** If $t \in \mathcal{B}$, then we have $\eta_t \geq \alpha\beta/(2M_1 + \phi(t))$.

*Proof.* If $t \in \mathcal{B}$, by Lemma 1 we can lower bound the step size $\eta_t$ by

$$\eta_t \geq \frac{\alpha\beta\|\tilde{\mathbf{x}}_t - \mathbf{x}_t\|}{\|\mathbf{g}(\tilde{\mathbf{x}}_t) - \mathbf{g}(\mathbf{x}_t) - \tilde{\mathbf{H}}_t(\tilde{\mathbf{x}}_t - \mathbf{x}_t)\|} = \frac{\alpha\beta}{\mathcal{E}_t}, \tag{32}$$

where $\mathcal{E}_t \triangleq \frac{\|\mathbf{g}(\tilde{\mathbf{x}}_t) - \mathbf{g}(\mathbf{x}_t) - \tilde{\mathbf{H}}_t(\tilde{\mathbf{x}}_t - \mathbf{x}_t)\|}{\|\tilde{\mathbf{x}}_t - \mathbf{x}_t\|}$ is the normalized approximation error. Moreover, as outlined in Section 3.1, we can apply the triangle inequality to upper bound $\mathcal{E}_t$. Specifically, we have

$$\mathcal{E}_t \leq \frac{\|\mathbf{g}(\tilde{\mathbf{x}}_t) - \mathbf{g}(\mathbf{x}_t) - \mathbf{H}_t(\tilde{\mathbf{x}}_t - \mathbf{x}_t)\|}{\|\tilde{\mathbf{x}}_t - \mathbf{x}_t\|} + \|\mathbf{H}_t - \bar{\mathbf{H}}_t\| + \|\bar{\mathbf{E}}_t\|. \tag{33}$$

By (9) in Lemma 2, the first term in (33) is upper bounded by $M_1$. Moreover, it also follows from Lemma 4 that $\|\mathbf{H}_t - \bar{\mathbf{H}}_t\| \leq \frac{1}{t+1}\sum_{i=0}^t \|\mathbf{H}_t - \mathbf{H}_i\| \leq M_1$. Hence, we further obtain $\mathcal{E}_t \leq 2M_1 + \|\bar{\mathbf{E}}_t\| \leq 2M_1 + \phi(t)$. Combining this with (32), we obtain the desired result. $\square$

Lemma 7 provides a lower bound on the step size $\eta_t$, but only for the case where $t \in \mathcal{B}$. In the next lemma, we further use induction to show a lower bound for the step sizes in all iterations.

**Lemma 8.** Assume that $\beta \leq \frac{1}{2}$. For any $t \geq 0$, we have

$$\eta_t \geq \min\left\{\frac{\alpha\beta}{2M_1 + \phi(t)}, \frac{\sigma_0}{\beta^t}\right\} \tag{34}$$

*Proof.* We prove this lemma by induction. For the base case $t = 0$, we consider two subcases. If $0 \in \mathcal{B}$, then by Lemma 7 we obtain that $\eta_0 \geq \frac{\alpha\beta}{2M_1 + \phi(0)}$. Otherwise, if $0 \notin \mathcal{B}$, we have $\eta_0 = \sigma_0$. In both cases, we observe that (34) is satisfied for the base case $t = 0$.

Now assume that (34) is satisfied for $t = s$ where $s \geq 0$. For $t = s + 1$, we again distinguish two subcases. If $s + 1 \in \mathcal{B}$, then by Lemma 7 we obtain that $\eta_{s+1} \geq \frac{\alpha\beta}{2M_1 + \phi(s+1)}$, which implies that (34) is satisfied. Otherwise, if $s + 1 \notin \mathcal{B}$, then we have

$$\eta_{s+1} = \sigma_{s+1} = \frac{\eta_s}{\beta} \geq \frac{1}{\beta}\min\left\{\frac{\alpha\beta}{2M_1 + \phi(s)}, \frac{\sigma_0}{\beta^s}\right\} = \min\left\{\frac{\alpha}{2M_1 + \phi(s)}, \frac{\sigma_0}{\beta^{s+1}}\right\}, \tag{35}$$

where we used the induction hypothesis in the last inequality. Furthermore, note that $\phi(s)/\phi(s+1) \leq \sqrt{\frac{s+2}{s+1}} \leq 2 \leq \frac{1}{\beta}$, which implies that $\phi(s) \leq \phi(s+1)/\beta$. Hence, we have

$$\frac{\alpha}{2M_1 + \phi(s)} \geq \frac{\alpha\beta}{2\beta M_1 + \phi(s+1)} \geq \frac{\alpha\beta}{2M_1 + \phi(s+1)}.$$

Therefore, (35) implies that $\eta_{s+1} \geq \min\left\{\frac{\alpha\beta}{2M_1 + \phi(s+1)}, \frac{\sigma_0}{\beta^{s+1}}\right\}$ and thus (34) also holds in this subcase. This completes the induction and we conclude that (34) holds for all $t \geq 0$. $\qquad\square$

As a corollary of Lemma 8, we obtain the following lower bound on $\eta_t$ for $t \geq \mathcal{T}_1$.

**Corollary 2.** *Recall the definition of $\mathcal{T}_1$ in (27). For any $t \geq \mathcal{T}_1$, we have $\eta_t \geq \alpha\beta/(3M_1)$.*

*Proof.* As shown in [1, Lemma 2], we have $\phi(t) \leq M_1$ when $t \geq \max\left\{256\frac{\Upsilon^2}{\kappa^2}\log\frac{8d\Upsilon}{\kappa\delta}, 4\log\frac{d}{\delta}\right\}$. Moreover, we have $\frac{\sigma_0}{\beta^t} \geq \frac{\alpha\beta}{3M_1}$ when $t \geq \log_{\frac{1}{\beta}}\frac{\alpha\beta}{3M_1\sigma_0}$. Hence, by Lemma 8 we conclude that $\eta_t \geq \frac{\alpha\beta}{3M_1}$ when $t \geq \mathcal{T}_1$. $\qquad\square$

Now we are ready to prove Lemma 6.

*Proof of Lemma 6.* By Proposition 1, we have $\|\mathbf{x}_{t+1} - \mathbf{x}^*\|^2 \leq \|\mathbf{x}_t - \mathbf{x}^*\|^2 (1 + 2\eta_t\mu)^{-1}$. By using Corollary 2, we obtain that $\|\mathbf{x}_{t+1} - \mathbf{x}^*\|^2 \leq \|\mathbf{x}_t - \mathbf{x}^*\|^2(1 + 2\alpha\beta\mu/(3M_1))^{-1} = \|\mathbf{x}_t - \mathbf{x}^*\|^2(1 + 2\alpha\beta/(3\kappa))^{-1}$. $\qquad\square$

## B.4 Proof of Theorem 3

In this section, we present the proof of Theorem 3. The proof consists of four steps. To begin with, in Lemma 9 we show that the iterate $\mathbf{x}_t$ stays in a local neighborhood of $\mathbf{x}^*$ when $t \geq \mathcal{I}$, where $\mathcal{I}$ is defined in (28). Next, we use this result in Lemma 10 to upper bound $\frac{1}{\sqrt{\mu\eta_t}}$ in those iterations where our line search scheme backtracks the step size, i.e., $t \in \mathcal{B}$. Then we use induction in Lemma 11 to prove an upper bound for all $t \geq 0$. Furthermore, we again use induction in Lemma 12 to establish an improved upper bound on $\frac{1}{\sqrt{\mu\eta_t}}$ when $t \geq \mathcal{T}_2$, where $\mathcal{T}_2$ is defined in (29). After proving Lemma 12, Theorem 3 then follows from Proposition 1.

**Lemma 9.** *We have $\|\mathbf{x}_t - \mathbf{x}^*\| \leq \nu\mu/L_2$ for all $t \geq \mathcal{I}$, where $\mathcal{I}$ is given in (28).*

*Proof.* By applying Lemma 6, we have $\|\mathbf{x}_{\mathcal{T}_1+u} - \mathbf{x}^*\|^2 \leq \|\mathbf{x}_{\mathcal{T}_1} - \mathbf{x}^*\|^2\left(1 + \frac{2\alpha\beta}{3\kappa}\right)^{-u}$. Moreover, since $\|\mathbf{x}_t - \mathbf{x}^*\|$ is non-increasing in $t$, we have $\|\mathbf{x}_{\mathcal{T}_1} - \mathbf{x}^*\| \leq \|\mathbf{x}_0 - \mathbf{x}^*\| = D$. Thus, we have

$$\|\mathbf{x}_{\mathcal{T}_1+u} - \mathbf{x}^*\| \leq \frac{\nu\mu}{L_2} \impliedby D^2\left(1 + \frac{2\alpha\beta}{3\kappa}\right)^{-u} \leq \frac{\nu^2\mu^2}{L_2^2} \impliedby u \geq 2\left(1 + \frac{3\kappa}{2\alpha\beta}\right)\log\left(\frac{L_2 D}{\nu\mu}\right).$$

This completes the proof. $\qquad\square$

Note that by Proposition 1, we have $\|\mathbf{x}_{t+1} - \mathbf{x}^*\|^2 \leq \|\mathbf{x}_t - \mathbf{x}^*\|^2(1 + 2\eta_t\mu)^{-1} \leq \|\mathbf{x}_t - \mathbf{x}^*\|^2/(2\eta_t\mu)$, which further implies that

$$\|\mathbf{x}_{t+1} - \mathbf{x}^*\| \leq \frac{\|\mathbf{x}_t - \mathbf{x}^*\|}{\sqrt{2\eta_t\mu}}. \tag{36}$$

Hence, to characterize the convergence rate of our method, it is sufficient to upper bound the quantity $1/\sqrt{2\eta_t\mu}$. We achieve this goal in the subsequent lemmas.

**Lemma 10.** *If $t \in \mathcal{B}$ and $t \geq \mathcal{I}$, then*

$$\frac{1}{\sqrt{2\eta_t\mu}} \leq \frac{L_2\|\mathbf{x}_t - \mathbf{x}^*\|}{4\alpha\beta\sqrt{(1-\alpha^2)\beta\mu}} + \frac{\|\bar{\mathbf{E}}_t\|}{2\alpha\sqrt{\beta\mu}} + \frac{3\kappa\mathcal{I}}{2\alpha\sqrt{\beta}(t+1)}. \tag{37}$$

*Moreover, it also holds that*

$$\frac{1}{\sqrt{2\eta_t\mu}} \leq \frac{\nu}{4\alpha\beta\sqrt{(1-\alpha^2)\beta}} + \frac{\phi(t)}{2\alpha\sqrt{\beta\mu}} + \frac{3\kappa\mathcal{I}}{2\alpha\sqrt{\beta}(t+1)}. \tag{38}$$

*Proof.* By using the second bound in Lemma 1, we obtain that

$$\frac{1}{\sqrt{2\eta_t\mu}} \leq \frac{\|\mathbf{g}(\tilde{\mathbf{x}}_t) - \mathbf{g}(\mathbf{x}_t) - \tilde{\mathbf{H}}_t(\tilde{\mathbf{x}}_t - \mathbf{x}_t)\|}{2\alpha\sqrt{\beta\mu}\|\tilde{\mathbf{x}}_t - \mathbf{x}_t\|} = \frac{\mathcal{E}_t}{2\alpha\sqrt{\beta\mu}}. \tag{39}$$

Furthermore, by combining (33) and (9) in Lemma 2, we further have

$$\mathcal{E}_t \leq \frac{L_2\|\mathbf{x}_t - \mathbf{x}^*\|}{2\beta\sqrt{1-\alpha^2}} + \|\mathbf{H}_t - \bar{\mathbf{H}}_t\| + \|\bar{\mathbf{E}}_t\|. \tag{40}$$

It remains to bound the bias term $\|\mathbf{H}_t - \bar{\mathbf{H}}_t\|$. By Lemma 4, we have

$$\|\mathbf{H}_t - \bar{\mathbf{H}}_t\| \leq \frac{1}{t+1}\sum_{i=0}^{t}\|\mathbf{H}_t - \mathbf{H}_i\| = \frac{1}{t+1}\sum_{i=0}^{\mathcal{I}-1}\|\mathbf{H}_t - \mathbf{H}_i\| + \frac{1}{t+1}\sum_{i=\mathcal{I}}^{t}\|\mathbf{H}_t - \mathbf{H}_i\|.$$

For $i = 0, 1\ldots,\mathcal{I}-1$, we use the first upper bound on $\|\mathbf{H}_t-\mathbf{H}_i\|$ in Lemma 4 to bound $\|\mathbf{H}_t-\mathbf{H}_i\| \leq M_1$, and thus $\sum_{i=0}^{\mathcal{I}-1}\|\mathbf{H}_t - \mathbf{H}_i\| \leq M_1\mathcal{I}$. Moreover, for $i = \mathcal{I},\mathcal{I}+1,\ldots,t$, we use the second upper bound in Lemma 4 to get $\|\mathbf{H}_t - \mathbf{H}_i\| \leq 2L_2\|\mathbf{x}_i - \mathbf{x}^*\|$. Moreover, note that $\mathbf{x}_i$ converges linearly to $\mathbf{x}^*$ when $i \geq \mathcal{I}$ by Lemma 6. Hence, we further have

$$\frac{1}{t+1}\sum_{i=\mathcal{I}}^{t}\|\mathbf{H}_t - \mathbf{H}_i\| \leq \frac{2L_2}{t+1}\sum_{i=\mathcal{I}}^{t}\|\mathbf{x}_i - \mathbf{x}^*\| \leq \frac{2L_2\|\mathbf{x}_\mathcal{I} - \mathbf{x}^*\|}{t+1}\sum_{i=0}^{\infty}\left(1 + \frac{2\alpha\beta}{3\kappa}\right)^{-i/2}$$
$$\leq \frac{4\nu\mu}{t+1}\left(1 + \frac{3\kappa}{2\alpha\beta}\right). \tag{41}$$

In the last inequality, we used the fact that $\|\mathbf{x}_\mathcal{I} - \mathbf{x}^*\| \leq \nu\mu/L_2$ and $\sum_{i=0}^{\infty}(1+\phi)^{-i/2} = 1/(1 - (1+\phi)^{-1/2}) = (1+\phi)^{1/2}/((1+\phi)^{1/2} - 1) = (1+\phi)^{1/2}((1+\phi)^{1/2} + 1)/\phi \leq 2(1 + 1/\phi)$, where $\phi = 2\alpha\beta/(3\kappa)$. Moreover, since $\mathcal{I} \geq 2\left(1 + \frac{3\kappa}{2\alpha\beta}\right)$ and $\nu \leq 1$, from (41) we further have $\frac{1}{t+1}\sum_{i=\mathcal{I}}^{t}\|\mathbf{H}_t - \mathbf{H}_i\| \leq \frac{2\mu\mathcal{I}}{t+1}$. Combining the above inequalities, we arrive at

$$\|\mathbf{H}_t - \bar{\mathbf{H}}_t\| \leq \frac{M_1\mathcal{I}}{t+1} + \frac{2\mu\mathcal{I}}{t+1} \leq \frac{3M_1\mathcal{I}}{t+1}. \tag{42}$$

Combining (39), (40), and (42) leads to the first result in (37). Finally, (38) follows from the fact that $\|\bar{\mathbf{E}}_t\| \leq \phi(t)$ and $\|\mathbf{x}_t - \mathbf{x}^*\| \leq \nu\mu/L_2$ for all $t \geq \mathcal{I}$. □

**Lemma 11.** *Assume that $\beta \leq 1/2$. For any $t \geq \mathcal{I}$, we have*

$$\frac{1}{\sqrt{2\eta_t\mu}} \leq \frac{\nu}{4\alpha\beta\sqrt{(1-\alpha^2)\beta}} + \frac{\phi(t)}{2\alpha\sqrt{\beta}\mu} + \frac{3\kappa\mathcal{I}}{2\alpha\sqrt{\beta}(t+1)} = \frac{\nu}{4\alpha\beta\sqrt{(1-\alpha^2)\beta}} + \rho_t, \tag{43}$$

*where $\rho_t$ is defined in* (30).

*Proof.* We shall use induction to prove Lemma 11. For $t = \mathcal{I}$, note that by Corollary 2, we have $\eta_\mathcal{I} \geq \alpha\beta/(3M_1)$. Thus, this implies that

$$\frac{1}{\sqrt{2\eta_\mathcal{I}\mu}} \leq \frac{\sqrt{3\kappa}}{\sqrt{2\alpha\beta}} \leq \frac{3\kappa\mathcal{I}}{2\alpha\sqrt{\beta}(\mathcal{I}+1)},$$

where we used the fact that $\kappa \geq 1$, $\alpha < 1$, $\beta \leq 1/2$ and $\mathcal{I} \geq 4$. This proves the base case where $t = \mathcal{I}$.

Now assume that (43) holds for $t = s$, where $s \geq \mathcal{I}$. For $t = s + 1$, we distinguish two subcases. If $s + 1 \in \mathcal{B}$, then by Lemma 10 we obtain that (43) is satisfied for $t = s + 1$. Otherwise, if $s + 1 \notin \mathcal{B}$, then we have $\eta_{s+1} = \sigma_{s+1} = \eta_s/\beta$. Hence, by using the induction hypothesis, we have

$$\frac{1}{\sqrt{2\eta_{s+1}\mu}} = \frac{\sqrt{\beta}}{\sqrt{2\eta_s\mu}} \leq \frac{\nu}{4\alpha\beta\sqrt{(1-\alpha^2)\beta}} + \frac{\sqrt{\beta}\phi(s)}{2\alpha\sqrt{\beta}\mu} + \frac{3\sqrt{\beta}\kappa\mathcal{I}}{2\alpha\sqrt{\beta}(s+1)}.$$

Since $\beta \leq 1/2$ and $\mathcal{I} \geq 2$, we have $(s + 2)/(s + 1) \leq (\mathcal{I} + 2)/(\mathcal{I} + 1) \leq 1.4 \leq 1/\sqrt{\beta}$ and $\phi(s) \leq \phi(s + 1)/\sqrt{\beta}$. Thus, we further have

$$\frac{1}{\sqrt{2\eta_{s+1}\mu}} \leq \frac{\nu}{4\alpha\beta\sqrt{(1-\alpha^2)\beta}} + \frac{\phi(s+1)}{2\alpha\sqrt{\beta}\mu} + \frac{3\kappa\mathcal{I}}{2\alpha\sqrt{\beta}(s+2)}.$$

This shows that (43) also holds in this subcase. This completes the induction and we conclude that (43) holds for all $t \geq \mathcal{I}$. □

Before proving Lemma 12, recall the definition of $\phi$ in (31) and first define

$$\mathcal{I}' = \sup_t \{t : \phi(t) \geq \nu\mu\} \quad \text{and} \quad \mathcal{T}_2' = \max\left\{\mathcal{I}', \frac{\kappa\mathcal{I}}{\nu} - 1\right\}. \tag{44}$$

Note that we have $\phi(t) \leq \nu\mu$ when $t \geq \frac{256\Upsilon^2}{\nu^2}\log\frac{8d\Upsilon}{\delta\nu}$. Hence, by the definition of (29), it holds that $\mathcal{T}_2 \geq \mathcal{T}_2'$.

**Lemma 12.** *Recall the definition of $\rho_t$ in (30). For any $t \geq \mathcal{T}_2'$, we have*

$$\frac{L_2\|\mathbf{x}_t - \mathbf{x}^*\|}{2\alpha\sqrt{\beta}\mu} \leq \rho_t \quad \text{and} \quad \frac{1}{\sqrt{2\mu\eta_t}} \leq \left(\frac{1}{2\beta\sqrt{1-\alpha^2}} + 5\right)\rho_t. \tag{45}$$

*Proof.* By Lemma 10, if $t \in \mathcal{B}$, then

$$\frac{1}{\sqrt{\mu\eta_t}} \leq \frac{L_2\|\mathbf{x}_t - \mathbf{x}^*\|}{2\alpha\beta\sqrt{2(1-\alpha^2)\beta}\mu} + \rho_t.$$

We shall prove (45) by induction. First consider the base case where $t = \mathcal{T}_2'$, where $\mathcal{T}_2'$ is defined in (44). To begin with, we will show that $\frac{\nu}{2\alpha\sqrt{\beta}} \leq \rho_{\mathcal{T}_2'} \leq \frac{5\nu}{2\alpha\sqrt{\beta}}$. Since $\mathcal{T}_2'$ is the maximum of $\mathcal{I}'$ and $\frac{\kappa\mathcal{I}}{\nu}$, we have either $\mathcal{T}_2' = \mathcal{I}'$ or $\mathcal{T}_2' = \frac{\kappa\mathcal{I}}{\nu} - 1$. In the former case, we can lower bound $\rho_{\mathcal{T}_2'} \geq \frac{1}{2\alpha\sqrt{\beta}\mu}\phi(\mathcal{I}') \geq \frac{\nu}{2\alpha\sqrt{\beta}}$. In the latter case, we can lower bound $\rho_{\mathcal{T}_2'} \geq \frac{3\kappa\mathcal{I}}{2\alpha\sqrt{\beta}(\mathcal{T}_2'+1)} = \frac{3\nu}{2\alpha\sqrt{\beta}}$. Combining both cases leads to the lower bound on $\rho_{\mathcal{T}_2'}$. Furthermore, note that both two terms in $\rho_t$ are a decreasing function in terms of $t$, and hence we have

$$\rho_{\mathcal{T}_2'} \leq \frac{1}{2\alpha\sqrt{\beta}\mu}\phi(\mathcal{I}') + \frac{3\kappa\mathcal{I}}{2\alpha\sqrt{\beta}(\kappa\mathcal{I}/\nu)} \leq \frac{2}{2\alpha\sqrt{\beta}\mu}\phi(\mathcal{I}'+1) + \frac{3\nu}{2\alpha\sqrt{\beta}} \leq \frac{5\nu}{2\alpha\sqrt{\beta}}.$$

This proves the upper bound on $\rho_{\mathcal{T}_2'}$.

Now we return to the proof in the base case where $t = \mathcal{T}_2'$. since $\|\mathbf{x}_{\mathcal{T}_2'} - \mathbf{x}^*\| \leq \nu\mu/L_2$ by Lemma 9, we obtain that $\frac{L_2\|\mathbf{x}_{\mathcal{T}_2'} - \mathbf{x}^*\|}{2\alpha\sqrt{\beta}\mu} \leq \frac{\nu}{2\alpha\sqrt{\beta}} \leq \rho_{\mathcal{T}_2'}$. Moreover, by Lemma 11, we have

$$\frac{1}{\sqrt{2\eta_{\mathcal{T}_2'}\mu}} \leq \frac{\nu}{4\alpha\beta\sqrt{(1-\alpha^2)\beta}} + \rho(\mathcal{T}_2') \leq \frac{\nu}{4\alpha\beta\sqrt{(1-\alpha^2)\beta}} + \frac{5\nu}{2\alpha\sqrt{\beta}} \leq \left(\frac{1}{2\beta\sqrt{1-\alpha^2}} + 5\right)\rho_{\mathcal{T}_2'}.$$

This shows that (45) holds for $t = \mathcal{T}_2'$.

Now assume that (45) holds for $t = s \geq \mathcal{T}_2'$. For $t = s + 1$, by using the induction hypothesis and (36), we obtain that

$$\frac{L_2\|\mathbf{x}_{s+1} - \mathbf{x}^*\|}{2\alpha\sqrt{\beta}\mu} \leq \frac{L_2\|\mathbf{x}_s - \mathbf{x}^*\|}{2\alpha\sqrt{\beta}\mu\sqrt{2\eta_s\mu}} \leq \frac{1}{\sqrt{2}}\left(\frac{1}{2\beta\sqrt{1-\alpha^2}} + 5\right)\rho_s^2.$$

Moreover, since $\rho_s/2 \leq \rho_{s+1}$, it suffices to show that $\frac{1}{\sqrt{2}}\left(\frac{1}{2\beta\sqrt{1-\alpha^2}} + 5\right)\rho_s^2 \leq \rho_s/2$, which is equivalent to $\sqrt{2}\left(\frac{1}{2\beta\sqrt{1-\alpha^2}} + 5\right)\rho_s \leq 1$. Furthermore, since $\rho_s$ is non-increasing, we further have

$$\sqrt{2}\left(\frac{1}{2\beta\sqrt{1-\alpha^2}} + 5\right)\rho_s \leq \sqrt{2}\left(\frac{1}{2\beta\sqrt{1-\alpha^2}} + 5\right)\rho_{\mathcal{T}_2'} \leq \sqrt{2}\left(\frac{1}{2\beta\sqrt{1-\alpha^2}} + 5\right)\frac{5\nu}{2\alpha\sqrt{\beta}} \leq 1,$$

where we used the condition on $\nu$ stated in Theorem 3 in the last inequality. This proves the first inequality in (45).

To prove the second inequality in (45) for $t = s + 1$, we distinguish two subcases. If $s + 1 \in \mathcal{B}$, then by Lemma 10, we have

$$\frac{1}{\sqrt{2\eta_{s+1}\mu}} \leq \frac{L_2\|\mathbf{x}_{s+1} - \mathbf{x}^*\|}{4\alpha\beta\sqrt{(1-\alpha^2)\beta}\mu} + \frac{\|\bar{\mathbf{E}}_{s+1}\|}{2\alpha\sqrt{\beta}\mu} + \frac{3\kappa\mathcal{I}}{2\alpha\sqrt{\beta}(s+1)} \leq \frac{1}{2\beta\sqrt{1-\alpha^2}}\rho_{s+1} + \rho_{s+1}$$

$$\leq \left(\frac{1}{2\beta\sqrt{1-\alpha^2}} + 5\right)\rho_{s+1}.$$

Otherwise, if $s + 1 \notin \mathcal{B}$, then we have $\eta_{s+1} = \eta_s/\beta$ and hence

$$\frac{1}{\sqrt{2\eta_{s+1}\mu}} = \frac{\sqrt{\beta}}{\sqrt{2\eta_s\mu}} \le \left(\frac{1}{2\beta\sqrt{1-\alpha^2}} + 5\right)\sqrt{\beta}\rho_s.$$

Since $\mathcal{T}_2' \ge \mathcal{I} \ge 2$ and $\beta \le 1/2$, we have $\rho_s/\rho_{s+1} \le (\mathcal{I}+2)/(\mathcal{I}+1) \le 1.4 \le 1/\sqrt{\beta}$. Thus, we also proved that $\frac{1}{\sqrt{\mu\eta_{s+1}}} \le \left(\frac{1}{2\beta\sqrt{1-\alpha^2}} + 5\right)\rho_{s+1}$. This completes the induction. □

*Proof of Theorem 3.* It immediately follows from (36) and Lemma 12. □

# C   Missing Proofs in Section 5

In this section, we will present the formal version of Theorem 2. Our proof largely mirrors the developments in Section 4.

## C.1   Approximation error analysis

**Averaged stochastic error.** Similar to Lemma 3, we can use tools from concentration inequalities to prove the following upper bound on the averaged stochastic error.

**Lemma 13** ([1, Lemma 6]). *Let $\delta \in (0,1)$ with $d/\delta \ge e$. Then with probability $1 - \delta\pi^2/6$, we have, for any $t \ge 0$,*

$$\|\bar{\mathbf{E}}_t\| \le 8\Psi\Upsilon_E \max\left\{\sqrt{\log\left(\frac{d(t+1)}{\delta}\right)\frac{w'(t)}{w(t)}}, \log\left(\frac{d(t+1)}{\delta}\right)\frac{w'(t)}{w(t)}\right\}. \tag{46}$$

## C.2   Convergence analysis

**Warm-up phase.** Similar to the case of uniform averaging, we can only ensure that the distance to $\mathbf{x}^*$ is monotonically non-increasing by Proposition 1 during this phase. Moreover, Algorithm 1 transitions to the linear phase when $\|\bar{\mathbf{E}}_t\| \le M_1$. Specifically, the transition point $\mathcal{U}_1$ is given by

$$\mathcal{U}_1 = \sup_t\left\{t \ge \log_{\frac{1}{\beta}}\frac{\alpha\beta}{3M_1\sigma_0} : \log\left(\frac{d(t+1)}{\delta}\right)\frac{w'(t)}{w(t)} \ge \left(1 \wedge \frac{\kappa}{8\Upsilon}\right)^2\right\} + 1. \tag{47}$$

When $w(t) = (t+1)^{\log(t+4)}$, we have $w'(t)/w(t) = \mathcal{O}\left(\log(t)/t\right)$. Thus, we conclude that $\mathcal{U}_1 = \tilde{\mathcal{O}}(\Upsilon^2/\kappa^2)$.

**Linear convergence phase**. In the following lemma, we prove the linear convergence of Algorithm 1 with weighted averaging.

**Lemma 14.** *Assume that $\beta \le 1/\Psi^2$ and recall the definition of $\mathcal{U}_1$ in (47). For any $t \ge \mathcal{U}_1$, we have*

$$\|\mathbf{x}_{t+1} - \mathbf{x}^*\|^2 \le \|\mathbf{x}_t - \mathbf{x}^*\|^2\left(1 + \frac{2\alpha\beta}{3\kappa}\right)^{-1},$$

*where $\kappa \triangleq M_1/\mu$ is the condition number.*

*Proof.* See Appendix C.3. □

**Superlinear convergence phase.** Define

$$\mathcal{J}' = \sup_t\left\{t : \log\left(\frac{d(t+1)}{\delta}\right)\frac{w'(t)}{w(t)} \ge \left(1 \wedge \frac{1}{8\Upsilon}\right)^2\right\} + 1. \tag{48}$$

Moreover, let $\nu \in (0,1)$ be a parameter and define

$$\mathcal{J} = \max\left\{\mathcal{U}_1 + 2\left(1 + \frac{2\kappa}{\alpha\beta}\right)\log\frac{L_2 D}{\nu\mu}, \mathcal{J}_2'\right\}. \tag{49}$$

Finally, let

$$\mathcal{U}_2 = \sup_t \left\{ t : w(t) \leq w(\mathcal{J}) \frac{\kappa}{\nu} \right\}. \tag{50}$$

When $w(t) = (t+1)^{\log(t+4)}$, we remark that $\mathcal{J}' = \tilde{\mathcal{O}}(\Upsilon^2)$ and thus $\mathcal{J} = \tilde{\mathcal{O}}(\kappa + \Upsilon^2)$. Moreover, similar to the derivation in [1], it can be shown that $\mathcal{U}_2 = \mathcal{O}(\mathcal{J}) = \tilde{\mathcal{O}}(\kappa + \Upsilon^2)$.

**Theorem 4.** *Let $\nu \in (0, 1)$ be a parameter satisfying*

$$\left( \frac{1}{2\alpha\beta\sqrt{(1-\alpha^2)\beta}} + \frac{5}{\alpha\sqrt{\beta}} \right) \nu \leq \frac{1}{\Psi}, \tag{51}$$

*and recall the definition of $\mathcal{U}_2$ in (50). For any $t \geq \mathcal{U}_2$, we have*

$$\|\mathbf{x}_{t+1} - \mathbf{x}^*\| \leq \left( \frac{1}{10\beta\sqrt{2(1-\alpha^2)}} + \frac{1}{\sqrt{2}} \right) \theta_t \|\mathbf{x}_t - \mathbf{x}^*\|,$$

*where*

$$\theta_t = \frac{8\Psi\Upsilon_E}{\alpha\sqrt{2\beta}\mu} \sqrt{\log\left( \frac{d(t+1)}{\delta} \right) \frac{w'(t)}{w(t)}} + \frac{5\kappa w(\mathcal{J})}{\alpha\sqrt{2\beta}w(t)}. \tag{52}$$

*Proof.* See Appendix C.4. ∎

### C.3  Proof of Lemma 14

To simplify the notation, define the function

$$\phi(t) = 8\Psi\Upsilon_E \max\left\{ \sqrt{\log\left( \frac{d(t+1)}{\delta} \right) \frac{w'(t)}{w(t)}}, \log\left( \frac{d(t+1)}{\delta} \right) \frac{w'(t)}{w(t)} \right\}.$$

Then we can rewrite (46) as $\|\bar{\mathbf{E}}_t\| \leq \phi(t)$ for all $t \geq 0$. Similar to Lemma 7, we have the following result.

**Lemma 15.** *If $t \in \mathcal{B}$, then we have $\eta_t \geq \alpha\beta/(2M_1 + \|\bar{\mathbf{E}}_t\|) \geq \alpha\beta/(2M_1 + \phi(t))$.*

**Lemma 16.** *Assume that $\beta \leq 1/\Psi$. For any $t \geq 0$, we have*

$$\eta_t \geq \min\left\{ \frac{\alpha\beta}{2M_1 + \phi(t)}, \frac{\sigma_0}{\beta^t} \right\} \tag{53}$$

*Proof.* We prove this lemma by induction. For $t = 0$, we distinguish two subcases. If $0 \in \mathcal{B}$, then by Lemma 7 we obtain that $\eta_0 \geq \frac{\alpha\beta}{2M_1 + \phi(0)}$. Otherwise, if $0 \notin \mathcal{B}$, we have $\eta_0 = \sigma_0$. In both cases, we observe that (53) is satisfied for the base case $t = 0$.

Now assume that (53) is satisfied for $t = s$. For $t = s + 1$, we again distinguish two subcases. If $s + 1 \in \mathcal{B}$, then by Lemma 15 we obtain that $\eta_{s+1} \geq \frac{\alpha\beta}{2M_1 + \phi(s+1)}$, which implies that (53) is satisfied. Otherwise, if $s + 1 \notin \mathcal{B}$, then we have

$$\eta_{s+1} = \sigma_{s+1} = \frac{\eta_s}{\beta} \geq \frac{1}{\beta} \min\left\{ \frac{\alpha\beta}{M_1 + \phi(s)}, \frac{\sigma_0}{\beta^s} \right\} = \min\left\{ \frac{\alpha}{M_1 + \phi(s)}, \frac{\sigma_0}{\beta^{s+1}} \right\}, \tag{54}$$

where we used the induction hypothesis in the last inequality. Furthermore, note that

$$\frac{\phi(s)}{\phi(s+1)} \leq \frac{w'(s)w(s+1)}{w'(s+1)w(s)} \leq \Psi \leq \frac{1}{\beta},$$

and hence

$$\frac{\alpha}{M_1 + \phi(s)} \geq \frac{\alpha\beta}{\beta M_1 + \phi(s+1)} \geq \frac{\alpha\beta}{M_1 + \phi(s+1)}.$$

Therefore, (54) implies that $\eta_{s+1} \geq \min\left\{ \frac{\alpha\beta}{M_1 + \phi(s+1)}, \frac{\sigma_0}{\beta^{s+1}} \right\}$ and thus (53) also holds in this case. This completes the induction and we conclude that (53) holds for all $t \geq 0$. ∎

**Corollary 3.** *Recall the definition of $\mathcal{U}_1$ in (47). For any $t \geq \mathcal{U}_1$, we have $\eta_t \geq \alpha\beta/(3M_1)$.*

*Proof.* By definition, we have $\mathcal{U}_1 \geq \log_{\frac{1}{\beta}} \frac{\alpha\beta}{3M_1\sigma_0}$, and thus $\frac{\sigma_0}{\beta^{\mathcal{U}_1}} \geq \frac{\alpha\beta}{3M_1}$. Moreover, we also have $\phi(t) \leq M_1$. Hence, by Lemma 8 we conclude that $\eta_t \geq \frac{\alpha\beta}{3M_1}$ when $t \geq \mathcal{U}_1$. $\qquad\square$

Now we are ready to prove Lemma 14.

*Proof of Lemma 14.* It follows from Proposition 1 and Corollary 3. $\qquad\square$

## C.4 Proof of Theorem 4

**Lemma 17.** *We have $\|\bar{\mathbf{E}}_t\| \leq \nu\mu$ and $\|\mathbf{x}_t - \mathbf{x}^*\| \leq \nu\mu/L_2$ for all $t \geq \mathcal{J}$.*

*Proof.* This follows from Lemmas 13 and 14. $\qquad\square$

**Lemma 18.** *If $t \in \mathcal{B}$ and $t \geq \mathcal{J}$, then*

$$\frac{1}{\sqrt{\mu\eta_t}} \leq \frac{L_2\|\mathbf{x}_t - \mathbf{x}^*\|}{2\alpha\beta\sqrt{2(1-\alpha^2)\beta\mu}} + \frac{\|\bar{\mathbf{E}}_t\|}{\alpha\sqrt{2\beta}\mu} + \frac{\kappa w(\mathcal{J}-1)}{\alpha\sqrt{2\beta}w(t)} + \frac{2\nu}{\alpha\sqrt{2\beta}} \tag{55}$$

*and also*

$$\frac{1}{\sqrt{\mu\eta_t}} \leq \frac{\nu}{2\alpha\beta\sqrt{2(1-\alpha^2)\beta}} + \frac{3\nu}{\alpha\sqrt{2\beta}} + \frac{\kappa w(\mathcal{J}-1)}{\alpha\sqrt{2\beta}w(t)}. \tag{56}$$

*Proof.* Similar to the proof in Lemma 10, note that

$$\frac{1}{\sqrt{\mu\eta_t}} \leq \frac{L_2\|\mathbf{x}_t - \mathbf{x}^*\|}{2\alpha\beta\sqrt{2(1-\alpha^2)\beta\mu}} + \frac{\|\mathbf{H}_t - \bar{\mathbf{H}}_t\|}{\alpha\sqrt{2\beta}\mu} + \frac{\|\bar{\mathbf{E}}_t\|}{\alpha\sqrt{2\beta}\mu}.$$

For the second term, note that

$$\bar{\mathbf{H}}_t = \sum_{i=0}^{t} z_{i,t}\mathbf{H}_i, \quad \text{where } z_{i,t} = \frac{w(i) - w(i-1)}{w(t)}.$$

Hence, by Jensen's inequality, we have

$$\|\mathbf{H}_t - \bar{\mathbf{H}}_t\| \leq \sum_{i=0}^{t} z_{i,t}\|\mathbf{H}_t - \mathbf{H}_i\| = \sum_{i=0}^{\mathcal{J}-1} z_{i,t}\|\mathbf{H}_t - \mathbf{H}_i\| + \sum_{i=\mathcal{J}}^{t} z_{i,t}\|\mathbf{H}_t - \mathbf{H}_i\|.$$

When $0 \leq i \leq \mathcal{J} - 1$, we use Assumption 2 to bound $\|\mathbf{H}_t - \mathbf{H}_i\| \leq M_1$ and thus

$$\sum_{i=0}^{\mathcal{J}-1} z_{i,t}\|\mathbf{H}_t - \mathbf{H}_i\| \leq M_1 \sum_{i=0}^{\mathcal{J}-1} \frac{w(i) - w(i-1)}{w(t)} = M_1 \frac{w(\mathcal{J}-1)}{w(t)}.$$

Moreover, for $\mathcal{J} \leq i \leq t$, we use Assumption 3 to get

$$\|\mathbf{H}_t - \mathbf{H}_i\| \leq L_2\|\mathbf{x}_t - \mathbf{x}_i\| \leq L_2\left(\|\mathbf{x}_t - \mathbf{x}^*\| + \|\mathbf{x}_i - \mathbf{x}^*\|\right) \leq 2L_2\|\mathbf{x}_i - \mathbf{x}^*\| \leq 2\nu\mu.$$

Thus, $\sum_{i=\mathcal{J}}^{t} z_{i,t}\|\mathbf{H}_t - \mathbf{H}_i\| \leq 2\nu\mu \sum_{i=\mathcal{I}}^{t} z_{i,t} \leq 2\nu\mu$.

Combining the above inequalities, we arrive at

$$\|\mathbf{H}_t - \bar{\mathbf{H}}_t\| \leq M_1 \frac{w(\mathcal{J}-1)}{w(t)} + 2\nu\mu.$$

This leads to the first result in (55). To show (56), we note that $\|\bar{\mathbf{E}}_t\| \leq \phi(t)$ and $\|\mathbf{x}_t - \mathbf{x}^*\| \leq \nu\mu/L_2$ for all $t \geq \mathcal{J}$. $\qquad\square$

**Lemma 19.** *Assume that $\beta \leq 1/\Psi^2$. For any $t \geq \mathcal{J}$, we have*

$$\frac{1}{\sqrt{\mu\eta_t}} \leq \frac{\nu}{2\alpha\beta\sqrt{2(1-\alpha^2)\beta}} + \frac{3\nu}{\alpha\sqrt{2\beta}} + \frac{\sqrt{2}\kappa w(\mathcal{J})}{\alpha\sqrt{\beta}w(t)}. \tag{57}$$

*Proof.* We shall use induction to prove Lemma 19. For $t = \mathcal{J}$, note that by Corollary 3, we have $\eta_{\mathcal{J}} \geq \alpha\beta/(3M_1)$. Thus, this implies that

$$\frac{1}{\sqrt{\mu\eta_{\mathcal{J}}}} \leq \frac{\sqrt{2\kappa}}{\sqrt{\alpha\beta}} \leq \frac{\sqrt{2}\kappa}{\alpha\sqrt{\beta}},$$

where we used $\kappa \geq 1$, $\alpha, \beta < 1$ and $\mathcal{I} \geq 2$. This proves the base case where $t = \mathcal{J}$.

Now assume that (57) holds for $t = s$, where $s \geq \mathcal{J}$. For $t = s + 1$, we distinguish two cases. If $s + 1 \in \mathcal{B}$, then by Lemma 18 we obtain that (57) is satisfied for $t = s + 1$. Otherwise, if $s + 1 \notin \mathcal{B}$, then we have $\eta_{s+1} = \sigma_{s+1} = \eta_s/\beta$. Hence, by using the induction hypothesis, we have

$$\frac{1}{\sqrt{\mu\eta_{s+1}}} = \frac{\sqrt{\beta}}{\sqrt{\mu\eta_s}} \leq \frac{\nu}{2\alpha\beta\sqrt{2(1-\alpha^2)\beta}} + \frac{3\nu}{\alpha\sqrt{2\beta}} + \frac{\sqrt{2\beta}\kappa w(\mathcal{J})}{\alpha\sqrt{\beta}w(s)}.$$

Since $\beta \leq 1/\Psi^2$, we have $w(s+1)/w(s) \leq \Psi \leq 1/\sqrt{\beta}$. Thus, we further have

$$\frac{1}{\sqrt{\mu\eta_{s+1}}} \leq \frac{\nu}{2\alpha\beta\sqrt{2(1-\alpha^2)\beta}} + \frac{3\nu}{\alpha\sqrt{2\beta}} + \frac{\sqrt{2}\kappa w(\mathcal{J})}{\alpha\sqrt{\beta}w(s+1)}.$$

This shows that (57) also holds in this case. $\quad\square$

Recall that $w(\mathcal{U}_2) = w(\mathcal{J})\frac{\kappa}{\nu}$. Then by Lemma 19, for $t \geq \mathcal{U}_2$ we have

$$\frac{1}{\sqrt{\mu\eta_t}} \leq \frac{\nu}{2\alpha\beta\sqrt{2(1-\alpha^2)\beta}} + \frac{3\nu}{\alpha\sqrt{2\beta}} + \frac{\sqrt{2}\kappa w(\mathcal{J})}{\alpha\sqrt{\beta}w(\mathcal{U}_2)} \leq \frac{\nu}{2\alpha\beta\sqrt{2(1-\alpha^2)\beta}} + \frac{3\nu}{\alpha\sqrt{2\beta}} + \frac{\sqrt{2}\nu}{\alpha\sqrt{\beta}}$$

$$= \frac{\nu}{2\alpha\beta\sqrt{2(1-\alpha^2)\beta}} + \frac{5\nu}{\alpha\sqrt{2\beta}}.$$

We will choose $\nu$ such that

$$\left(\frac{1}{2\alpha\beta\sqrt{(1-\alpha^2)\beta}} + \frac{5}{\alpha\sqrt{\beta}}\right)\nu \leq \frac{1}{\Psi}. \tag{58}$$

Therefore, this further implies that $\|\mathbf{x}_{t+1} - \mathbf{x}^*\| \leq \|\mathbf{x}_t - \mathbf{x}^*\|/(2\Psi)$ for $t \geq \mathcal{U}_2$.

**Lemma 20.** *If $t \in \mathcal{B}$ and $t \geq \mathcal{U}_2$, then*

$$\frac{1}{\sqrt{\mu\eta_t}} \leq \frac{L_2\|\mathbf{x}_t - \mathbf{x}^*\|}{2\alpha\beta\sqrt{2(1-\alpha^2)\beta\mu}} + \frac{\phi(t)}{\alpha\sqrt{2\beta\mu}} + \frac{5\kappa w(\mathcal{J})}{\alpha\sqrt{2\beta}w(t)}. \tag{59}$$

*Proof.* Recall that we have

$$\frac{1}{\sqrt{\mu\eta_t}} \leq \frac{L_2\|\mathbf{x}_t - \mathbf{x}^*\|}{2\alpha\beta\sqrt{2(1-\alpha^2)\beta\mu}} + \frac{\|\mathbf{H}_t - \bar{\mathbf{H}}_t\|}{\alpha\sqrt{2\beta\mu}} + \frac{\|\bar{\mathbf{E}}_t\|}{\alpha\sqrt{2\beta\mu}}.$$

For the second term, we can write

$$\|\mathbf{H}_t - \bar{\mathbf{H}}_t\| \leq \sum_{i=0}^{t} z_{i,t}\|\mathbf{H}_t - \mathbf{H}_i\| = \sum_{i=0}^{\mathcal{J}-1} z_{i,t}\|\mathbf{H}_t - \mathbf{H}_i\| + \sum_{i=\mathcal{J}}^{\mathcal{U}_2} z_{i,t}\|\mathbf{H}_t - \mathbf{H}_i\| + \sum_{i=\mathcal{U}_2+1}^{t} z_{i,t}\|\mathbf{H}_t - \mathbf{H}_i\|.$$

For the first part, $\sum_{i=0}^{\mathcal{J}-1} z_{i,t}\|\mathbf{H}_t - \mathbf{H}_i\| \leq M_1\frac{w(\mathcal{J}-1)}{w(t)} \leq M_1\frac{w(\mathcal{J})}{w(t)}$. For the second part,

$$\sum_{i=\mathcal{J}}^{\mathcal{U}_2} z_{i,t}\|\mathbf{H}_t - \mathbf{H}_i\| \leq 2\nu\mu\sum_{i=\mathcal{J}}^{\mathcal{U}_2} z_{i,t} \leq 2\nu\mu\frac{w(\mathcal{U}_2)}{w(t)} = 2M_1\frac{w(\mathcal{J})}{w(t)},$$

where we used the fact that $w(\mathcal{U}_2) = w(\mathcal{J})\frac{\kappa}{\nu}$. For the third part,

$$
\begin{aligned}
\sum_{i=\mathcal{U}_2+1}^{t} z_{i,t}\|\mathbf{H}_t - \mathbf{H}_i\| &\leq \sum_{i=\mathcal{U}_2+1}^{t} 2L_2 \frac{w(i) - w(i-1)}{w(t)}\|\mathbf{x}_i - \mathbf{x}^*\| \\
&\leq \sum_{j=1}^{t-\mathcal{U}_2} 2L_2 \frac{w(\mathcal{U}_2 + j) - w(\mathcal{U}_2 + j - 1)}{w(t)} \frac{\|\mathbf{x}_{\mathcal{U}_2} - \mathbf{x}^*\|}{(2\Psi)^j} \\
&\leq \sum_{j=1}^{t-\mathcal{U}_2} 2L_2 \frac{w(\mathcal{U}_2 + j)}{w(t)} \frac{\|\mathbf{x}_{\mathcal{U}_2} - \mathbf{x}^*\|}{(2\Psi)^j} \\
&\leq \sum_{j=1}^{t-\mathcal{U}_2} 2\nu\mu \frac{w(\mathcal{U}_2)}{w(t)} \frac{w(\mathcal{U}_2 + j)}{w(\mathcal{U}_2)(2\Psi)^j} \\
&\leq 2\nu\mu \frac{w(\mathcal{U}_2)}{w(t)} \sum_{j=1}^{t} \frac{w(\mathcal{U}_2 + j)}{w(\mathcal{U}_2)(2\Psi)^j} \\
&\leq 2\nu\mu \frac{w(\mathcal{U}_2)}{w(t)} \sum_{j=1}^{t} \frac{1}{2^j} \\
&\leq 2\nu\mu \frac{w(\mathcal{U}_2)}{w(t)} = \frac{2M_1 w(\mathcal{J})}{w(t)}.
\end{aligned}
$$

Therefore, we conclude that

$$
\|\mathbf{H}_t - \bar{\mathbf{H}}_t\| \leq \frac{3M_1 w(\mathcal{J})}{w(t)} + \frac{2M_1 w(\mathcal{J})}{w(t)} = \frac{5M_1 w(\mathcal{J})}{w(t)}.
$$

This completes the proof. $\qquad\square$

**Lemma 21.** *Recall the definition of $\theta_t$ in (52). For any $t \geq 0$, we have*

$$
\frac{5L_2\|\mathbf{x}_{\mathcal{U}_2+t} - \mathbf{x}^*\|}{\alpha\sqrt{2\beta}\mu} \leq \theta_t \quad \text{and} \quad \frac{1}{\sqrt{\mu\eta_{\mathcal{U}_2+t}}} \leq \left(\frac{1}{10\beta\sqrt{1-\alpha^2}} + 1\right)\theta_t, \tag{60}
$$

*Proof.* Note that by Proposition 1, we have

$$
\|\mathbf{x}_{\mathcal{U}_2+t+1} - \mathbf{x}^*\| \leq \|\mathbf{x}_{\mathcal{U}_2+t} - \mathbf{x}^*\|(1 + 2\eta_{\mathcal{U}_2+t}\mu)^{-1/2} \leq \frac{\|\mathbf{x}_{\mathcal{U}_2+t} - \mathbf{x}^*\|}{\sqrt{2\eta_{\mathcal{U}_2+t}\mu}}.
$$

By Lemma 18, if $\mathcal{U}_2 + t \in \mathcal{B}$, then

$$
\frac{1}{\sqrt{\mu\eta_{\mathcal{U}_2+t}}} \leq \frac{L_2\|\mathbf{x}_{\mathcal{U}_2+t} - \mathbf{x}^*\|}{2\alpha\beta\sqrt{2(1-\alpha^2)\beta}\mu} + \theta_t.
$$

We will prove (60) by induction. First consider the base case $t = 0$. We note that $\theta_0 \geq \frac{5\kappa w(\mathcal{J})}{\alpha\sqrt{2\beta}w(\mathcal{U}_2)} = \frac{5\nu}{\alpha\sqrt{2\beta}}$. On the other hand, since $\|\mathbf{x}_{\mathcal{U}_2} - \mathbf{x}^*\| \leq \nu\mu/L_2$, we obtain that $\frac{5L_2\|\mathbf{x}_{\mathcal{U}_2} - \mathbf{x}^*\|}{\alpha\sqrt{2\beta}\mu} \leq \frac{5\nu}{\alpha\sqrt{2\beta}} \leq \theta_0$. Moreover, by Lemma 19, we have

$$
\begin{aligned}
\frac{1}{\sqrt{\mu\eta_{\mathcal{U}_2}}} &\leq \frac{\nu}{2\alpha\beta\sqrt{2(1-\alpha^2)\beta}} + \frac{3\nu}{\alpha\sqrt{2\beta}} + \frac{\sqrt{2}\kappa w(\mathcal{J})}{\alpha\sqrt{\beta}w(\mathcal{U}_2)} \leq \frac{\nu}{2\alpha\beta\sqrt{2(1-\alpha^2)\beta}} + \frac{5\nu}{\alpha\sqrt{2\beta}} \\
&\leq \left(\frac{1}{10\beta\sqrt{1-\alpha^2}} + 1\right)\theta_0.
\end{aligned}
$$

This shows that (60) holds for $t = 0$.

Now assume that (60) holds for $t = s \geq 0$. For $t = s + 1$, by using the induction hypothesis, we obtain

$$
\frac{5L_2\|\mathbf{x}_{\mathcal{U}_2+s+1} - \mathbf{x}^*\|}{\alpha\sqrt{2\beta}\mu} \leq \frac{5L_2\|\mathbf{x}_{\mathcal{U}_2+s} - \mathbf{x}^*\|}{\alpha\sqrt{2\beta}\mu\sqrt{2\eta_{\mathcal{U}_2+s}\mu}} \leq \frac{1}{\sqrt{2}}\left(\frac{1}{10\beta\sqrt{1-\alpha^2}} + 1\right)\theta_s^2.
$$

Note that $\theta_s/\Psi \le \theta_{s+1}$. Thus, it suffices to show that $\frac{1}{\sqrt{2}}\left(\frac{1}{10\beta\sqrt{1-\alpha^2}}+1\right)\theta_s^2 \le \theta_s/\Psi$, which is equivalent to $\frac{\Psi}{\sqrt{2}}\left(\frac{1}{10\beta\sqrt{1-\alpha^2}}+1\right)\theta_s \le 1$. Furthermore, note that $\theta_s$ is non-increasing and $\theta_0 \le \frac{\nu}{\alpha\sqrt{2\beta}}+\frac{5\nu}{\alpha\sqrt{2\beta}}=\frac{6\nu}{\alpha\sqrt{2\beta}}$. Thus, we only need to require

$$\frac{\Psi}{\sqrt{2}}\left(\frac{1}{10\beta\sqrt{1-\alpha^2}}+1\right)\frac{6\nu}{\alpha\sqrt{2\beta}}\le 1 \quad\Leftrightarrow\quad \left(\frac{3}{10\alpha\beta\sqrt{(1-\alpha^2)\beta}}+\frac{3}{\alpha\sqrt{\beta}}\right)\nu\le\frac{1}{\Psi},$$

which is satisfied due to (51). This proves the first inequality in (60).

To prove the second inequality in (60) for $t=s+1$, we distinguish two cases. If $s+1\in\mathcal{B}$, then by Lemma 20, we have

$$\frac{1}{\sqrt{\mu\eta_{\mathcal{U}_2+s+1}}} \le \frac{L_2\|\mathbf{x}_{\mathcal{U}_2+s+1}-\mathbf{x}^*\|}{2\alpha\beta\sqrt{2(1-\alpha^2)\beta\mu}}+\frac{\|\bar{\mathbf{E}}_{\mathcal{U}_2+s+1}\|}{\alpha\sqrt{2\beta\mu}}+\frac{5\kappa w(\mathcal{J})}{\alpha\sqrt{2\beta}w(\mathcal{U}_2+s+1)}$$

$$\le \frac{1}{10\beta\sqrt{1-\alpha^2}}\theta_{s+1}+\theta_{s+1} \le \left(\frac{1}{10\beta\sqrt{1-\alpha^2}}+1\right)\theta_{s+1}.$$

Otherwise, if $s+1\notin\mathcal{B}$, then we have $\eta_{\mathcal{U}_2+s+1}=\eta_{\mathcal{U}_2+s}/\beta$ and hence

$$\frac{1}{\sqrt{\mu\eta_{\mathcal{U}_2+s+1}}}=\frac{\sqrt{\beta}}{\sqrt{\mu\eta_{\mathcal{U}_2+s}}}\le\left(\frac{1}{10\beta\sqrt{1-\alpha^2}}+1\right)\sqrt{\beta}\theta_s.$$

Since $\theta_s/\Psi\le\theta_{s+1}$ and $\sqrt{\beta}\le 1/\Psi$, this implies that $\frac{1}{\sqrt{\mu\eta_{\mathcal{T}_2+s+1}}}\le\left(\frac{1}{10\beta\sqrt{1-\alpha^2}}+1\right)\theta_{s+1}$. This completes the induction. $\qquad\square$

*Proof of Theorem 4.* By Proposition 1, we have

$$\|\mathbf{x}_{\mathcal{T}_2+t+1}-\mathbf{x}^*\|\le\|\mathbf{x}_{\mathcal{T}_2+t}-\mathbf{x}^*\|(1+2\eta_{\mathcal{T}_2+t}\mu)^{-1/2}\le\frac{\|\mathbf{x}_{\mathcal{T}_2+t}-\mathbf{x}^*\|}{\sqrt{2\eta_{\mathcal{T}_2+t}\mu}}.$$

The rest follows from Lemma 21. $\qquad\square$

## D  Additional discussions

### D.1  Iteration complexity of SNPE

Note that for $t\ge\mathcal{U}_3=\tilde{\mathcal{O}}(\Upsilon^2+\kappa)$, we have $\|\mathbf{x}_{t+1}-\mathbf{x}^*\|=\tilde{\mathcal{O}}(\frac{\Upsilon}{\sqrt{t}})\|\mathbf{x}_t-\mathbf{x}^*\|$ by Theorem 2. By unrolling the inequality, this implies that

$$\|\mathbf{x}_{t+1}-\mathbf{x}^*\|\le\|\mathbf{x}_{\mathcal{U}_3}-\mathbf{x}^*\|\tilde{\mathcal{O}}\left(\prod_{s=\mathcal{U}_3}^t\frac{\Upsilon}{\sqrt{s}}\right)\le\|\mathbf{x}_0-\mathbf{x}^*\|\tilde{\mathcal{O}}\left(\prod_{s=\mathcal{U}_3}^t\frac{\Upsilon}{\sqrt{s}}\right).$$

Further, for $t\ge 2\mathcal{U}_3$, we can upper bound $\frac{\Upsilon}{\sqrt{s}}$ as follows: (i) $\frac{\Upsilon}{\sqrt{s}}\le\frac{\Upsilon}{\sqrt{\mathcal{U}_3}}\le 1$ for any $s\in[\mathcal{U}_3,t/2]$, (ii) $\frac{\Upsilon}{\sqrt{s}}\le\frac{\Upsilon}{\sqrt{t/2}}$ for any $s\in[t/2,t]$. Thus, we have

$$\prod_{s=\mathcal{U}_3}^t\frac{\Upsilon}{\sqrt{s}}\le\prod_{s=t/2}^t\frac{\Upsilon}{\sqrt{s}}\le\left(\frac{\Upsilon}{\sqrt{t/2}}\right)^{\frac{t}{2}}.$$

To derive a complexity bound, we upper bound the required number of iterations $t$ such that $\left(\frac{\Upsilon}{\sqrt{t/2}}\right)^{\frac{t}{2}}=\epsilon$. Taking the logarithm of both sides and with some algebraic manipulation, we obtain

$$\frac{t}{2\Upsilon^2}\log\frac{t}{2\Upsilon^2}=\frac{2}{\Upsilon^2}\log\frac{1}{\epsilon}.$$

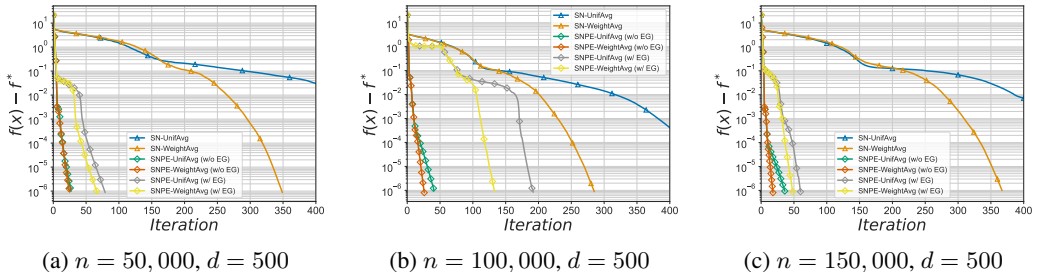

(a) $n = 50,000, d = 500$     (b) $n = 100,000, d = 500$     (c) $n = 150,000, d = 500$

Figure 3: The effect of the extragradient step in stochastic NPE.

Using the Lambert W function[1], the solution can be expressed as $\log \frac{t}{2\Upsilon^2} = W(\frac{2}{\Upsilon^2} \log \frac{1}{\epsilon}) \Rightarrow t = 2\Upsilon^2 e^{W(\frac{2}{\Upsilon^2} \log \frac{1}{\epsilon})}$. Finally, by applying the bound $e^{W(x)} \leq \frac{2x+1}{1+\log(x+1)}$ for any $x \geq 0$, we conclude that $t = \mathcal{O}\left(\frac{\log(\epsilon^{-1})}{\log(\Upsilon^{-2}\log(\epsilon^{-1}))}\right)$. Note that in the above derivation, we ignore the additional logarithmic factor $\log(t)$ in our superlinear convergence rate. However, a more careful analysis will show that it does not affect the final complexity bound. We also refer the reader to a similar derivation in [22, Appendix D.2], where the authors provide the same complexity bound for a similar convergence rate of $(1 + \mathcal{O}(\sqrt{t}))^{-t}$.

### D.2   The effect of the extragradient step

In Figure 3, we test the effect of the extragradient step in our proposed SNPE method. We observe that in all cases, the variant without an extragradient step outperforms the original version, suggesting that the extragradient step may not be beneficial for minimization problems. Nevertheless, the SNPE method with the extragradient step, which is the one analyzed in our paper, still outperforms the stochastic Newton method in [1].

---

[1]`https://en.wikipedia.org/wiki/Lambert_W_function`

