# OpenReview forum: "Stochastic Newton Proximal Extragradient Method"
_NeurIPS.cc/2024/Conference — NeurIPS 2024 poster_

### Official Review · Reviewer_TogL · 2024-07-05

**Soundness:** 3
**Presentation:** 3
**Contribution:** 3
**Rating:** 7
**Confidence:** 3

**Summary:**

The paper develops an accelerated scheme for strongly convex problems. As feedback it requires a deterministic first order oracles, but only an inexact Hessian estimator –– the main assumption being that the Hessian noise is mean zero and sub-exponential (implied by e.g. Hessian subsampling and $\Vert\nabla^2 f_i(x)\Vert$ bounded or Hessian sketching with a Gaussian sketch). The main idea is in combining the Hessian averaging scheme [1] with the hybrid proximal extragradient framework [18,19].

**Strengths:**

The paper is technically strong and very clearly/transparently written.

**Weaknesses:**

The main weaknesses seems to be that:

- It is unclear whether the scheme has relevance in practice, since the experiments seems to be run with a nonaccelerated version. For completeness I would suggest running the accelerated version, even if it performs worse (at least in the appendix). The theoretical results are strong and sufficient in themselves, so even though it would be better to find an instance that matches the theoretically guarantees, I don't see it as necessary.
- there are no guarantees for convex as also pointed out by the authors.

Neither are critical concerns – I mainly have a few remarks and questions regarding comparison with other methods and the experiments (see below).

**Questions:**

Method comparison & theory:

- It would be informative to compare theoretically with existing (accelerated) results using exact Hessian. Does the proposed method recover the existing complexities?
- l. 369 can you really claim better condition number dependency when you choose $x_{k+1}=\hat{x}_k$, which is not covered by the theory? What convergence guarantees can be shown then, since the interpolation in Eq. 4 otherwise appears crucial for the acceleration?
- How does the structure of the scheme compare with [1] when $x_{k+1}=\hat{x}_k$? It seems that the main difference is the stepsize selection (error condition vs Armijo style backtracking line search). If this is true, the experimental comparison can seem a bit synthetic.
- If you have full gradients, is it not possible to use Pearlmutter's implicit Hessian-vector product to (inexactly) do the second order update while still maintaining a $\mathcal{O}(nd)$ like first-order methods? See e.g. https://www.cs.toronto.edu/~jmartens/docs/Deep_HessianFree.pdf
- Is it possible to relax the requirement of exact gradients? For first order methods it is still possible to achieve exponential convergence under e.g. relative noise (see e.g. Thm 5.1 in https://arxiv.org/pdf/2102.02921).

Experimentally:

- What are choices of hyperparameters $\alpha, \beta$ of the method? How are the baselines tuned in comparison?
- The proposed method can also be used with a deterministic Hessian. How does the stochastic version compare with an exact Hessian? This would show the influence of the first (slow) phase and provide an idealized baseline.

Minor:

- Is it possible to shave off the logarithmic factors hiding in the $\widetilde{\mathcal O}$-notation?
- Figure 1: is it a labeling mistake or is SNPE-UnifAvg consistently better than SNPE-WeightAvg?
- How come stochastic NPE suffers an addition $\Upsilon$ dependency in the superlinear rate (in comparison with stochastic Newton does not)?

**Limitations:**

See weaknesses.

---

> ### Author Rebuttal · Authors · 2024-08-06
>
> **Q1 Running the accelerated version in the experiment.**
>
> **A1** We added numerical results for the original SNPE with the extragradient step and compared it with the variant without it (Figure 3 in the shared PDF). We observe that the modified variant outperforms the original, suggesting the extragradient step may not be beneficial. However, the original SNPE still outperforms the stochastic Newton method in [1].
>
> ---
> **Q2 No guarantees for convex.**
>
> **A2** Due to space limitations, please see our response to **Q1** for Reviewer **qDEj**.
>
> ---
> **Q3 The complexity of the proposed method with exact Hessians.**
>
> **A3** With exact Hessians, our method achieves a complexity of $O((\frac{L_2D}{\mu})^{2/3}+\log\log(1/\epsilon))$, though this requires a different analysis (we have an unpublished note). This matches the state-of-the-art complexity for second-order methods in strongly-convex strongly-concave min-max problems [R1]. However, for minimization problems, using Nesterov's acceleration achieves a better complexity of $O((\frac{L_2D}{\mu})^{2/7}+\log\log(1/\epsilon))$ [R2].
>
> [R1] Jiang, R. and Mokhtari, A. Generalized optimistic methods for convex-concave saddle point problems. (2022).
>
> [R2] Arjevani, Y. et al. Oracle complexity of second-order methods for smooth convex optimization. (2019).
>
> ---
> **Q4 Choosing $x_{k+1} = \hat{x}_k$ is not covered by the theory.**
>
> **A4** Good point. Our current theory only applies to SNPE in Algorithm 1 and cannot be easily extended to the modified version. However, drawing an analogy from first-order methods, it is reasonable to believe that both versions would share similar convergence guarantees. Specifically, the first-order instantiation of the HPE framework is the extragradient method, where the first step is $\hat{x}_t=x_t-\eta_t\nabla f(x_t)$. Since both extragradient and gradient descent achieve the same linear rate, it is plausible that the same analogy holds for SNPE and its modification. That said, we do not have concrete proof and leave this for future work.
>
> ---
> **Q5 Dffierence with stochastic Newton when $x_{k+1} = \hat{x}_k$?**
>
> **A5** Our modified scheme has a different update rule from stochastic Newton, in addition to the difference in step size selection. Our modified scheme is $x\_{t+1}=x\_t-\eta\_t(I+\eta_t\tilde{H}\_t)^{-1}\nabla f(x_t)$ as shown in Eq. (6). In contrast, stochastic Newton follows $x_{t+1}=x_t-\lambda_t\tilde{H}_t^{-1}\nabla f(x_t)$. Hence, the update directions are different, leading to distinct trajectories.
>
> ---
> **Q6 Using Pearlmutter's implicit Hessian-vector product for inexact second-order updates?**
>
> **A6** Good point. First, Pearlmutter's technique is not always applicable as it requires access to the computational graph of the objective. Moreover, in the Hessian-free methods, the number of Hessian-vector products per iteration can be substantial to maintain the superior convergence rate of the second-order update, often scaling with the (square root of the) condition number of the Hessian and the desired accuracy. Thus, the stochastic Hessian approximation can sometimes be more efficient for leveraging second-order information.
>
> ---
> **Q7 Relaxing the requirement of exact gradients.**
>
> **A7** Extending our analysis to noisy gradients is challenging since we cannot reliably check Condition (3), which is key in our proof. This limitation arises from the HPE framework and is not specific to our algorithm. Moreover, unless we impose strong assumptions on the gradient noise, it is unlikely to achieve superlinear convergence, which is our focus.
>
> ---
> **Q8 The choices of hyperparameters.**
>
> **A8** For our Algorithm 1, the hyperparameters are the line-search parameters $\alpha,\beta\in(0,1)$ and the initial step size $\sigma_0$. We chose $\alpha = 1,\beta = 0.5$, and $\sigma_0=1$ without optimizing their choices. For the stochastic Newton method, we followed the default line search parameters in the official GitHub implementation. Both algorithms are relatively robust to the choice of these hyerparameters. We will highlight this point in the revision.
>
> ---
> **Q9 The stochastic version v.s. an exact Hessian?**
>
> **A9** We included the numerical results for NPE (our method with exact Hessian) in the shared PDF; see Figures 1 and 2. Figure 1 shows that NPE achieves a faster superlinear convergence rate and converges in fewer iterations due to the use of the exact Hessians. However, our SNPE method also performs comparably to NPE, demonstrating the effectiveness of the Hessian averaging scheme. Moreover, in terms of runtime, SNPE outperforms NPE due to its lower per-iteration cost.
>
> ---
> **Q10 Logarithmic factors.**
>
> **A10** Eliminating the logarithmic factors seems difficult. These factors originate from Lemma 3, where we bound the average Hessian noise. The logarithmic dependence on $t$ arises from the union bound, while the dependence on $d$ is due to the matrix concentration inequality (Theorem 3 in [1]). Hence, these logarithmic factors cannot be eliminated without additional assumptions on the stochastic noise.
>
> ---
> **Q11 Figure 1: is it a labeling mistake?**
>
> **A11** It is not a labeling mistake; in this experiment, SNPE-UnifAvg is indeed better than SNPE-WeightAvg. This might be due to the small subsampling size, making the stochastic Hessian noise the limiting factor. In our new experiment in the shared PDF, we set the subsampling size to match the dimension, and we observed that the weighted averaging outperforms the uniform averaging in all cases.
>
> ---
> **Q12 An addition $\Upsilon$ dependency in stochastic NPE?**
>
> **A12** The transition points and superlinear rates of stochastic Newton also depend on $\Upsilon$. This dependence is not explicit in Table 1 because we focus on the setting where $\Upsilon = O(\kappa)$, which typically holds in practice. In this case, $O(\kappa^2 + \Upsilon^2) = O(\kappa^2)$, simplifying the expressions in Table 1. We will add a remark in the revision to clarity this.

---

> > ### Comment · Reviewer_TogL · 2024-08-09
> >
> > I thank the authors for their response and have no further questions.

---

### Official Review · Reviewer_BRp1 · 2024-07-12

**Soundness:** 2
**Presentation:** 2
**Contribution:** 2
**Rating:** 5
**Confidence:** 4

**Summary:**

This work proposes a novel algorithm called the Stochastic Newton Proximal Extragradient method. Authors claim that their method reaches a superlinear convergence rate after $\mathcal{O}(\kappa)$ iterations, in contrast to the $\mathcal{O}(\kappa^2)$ iterations proved in previous work.

**Strengths:**

- This paper proposes a new algorithm called the Stochastic Newton Proximal Extragradient method and proves its convergence.

**Weaknesses:**

- The assumptions are very restrictive. For example, assumptions 3 and 5 are very strong, and few machine learning applications can satisfy such assumptions.
- I am confused about how parts (c) and (d) of Theorem 1 prove superlinear convergence. From superlinear convergence, I expected to see $\rho^{2^t}$ where $\rho < 1$. How do the results show superlinear convergence?
- In Table 2, you mentioned the complexity of Damped Newton where the dependence on $\epsilon$ is $\log \log \epsilon$. However, for SNPE, the iteration complexity $\log \epsilon$ is similar to AGD. The iteration complexity of Damped Newton is called superlinear, while that of SNPE is called just linear.
- In lines 342- 344, the dominating term in the complexity is the one that depends on $\epsilon$. The Damped Newton has better dependence than SNPE.
- Section 7 needs more numerical experiments. I expect to see a comparison between SNPE and Damped Newton. In that comparison, putting **real-time (and not iterations)** on the $x$-axis will be fair, as SNPE does a line search in every iteration.
- In lines 334-337, the authors ignore the complexity of line-search in algorithm 1 (BLS). In the BLS, you compute $(1 + \eta \tilde{H})^{-1}g$ several times, which is very expensive, and authors ignore this while computing the complexity of SNPE.

**Questions:**

Check weaknesses.

---

> ### Author Rebuttal · Authors · 2024-08-06
>
> **Q1 The assumptions are very restrictive. e.g., Assumptions 3 and 5.**
>
> **A1** We note that our assumptions are standard in the study of (stochastic) second-order methods, including Subsampled Newton (Erdogdu & Montanari, 2015; Roosta-Khorasani & Mahoney, 2019), Newton Sketch (Pilanci & Wainwright, 2017; Agarwal et al., 2017), and notably, the recent work by Na et al. (2022).  In particular, Assumption 3 (Lipschitz Hessians) provides the necessary regularity condition for achieving superlinear convergence and is commonly used in the study of second-order methods, e.g., in Section 9.5.3 of the textbook by Boyd & Vandenberghe (2004). For instance, it is satisfied by the regularized log-sum-exp function and the loss function of regularized logistic regression. Moreover, we also discussed in Section 2 when Assumption 5 holds for stochastic Hessian approximation. Specifically, for Hessian subsampling, this is satisfied when each component function $f_i$ is convex, while it is automatically satisfied for Hessian sketching.
>
> ---
> **Q2 How do parts (c) and (d) of Theorem 1 prove superlinear convergence? I expected to see $\rho^{2^t}$ where $\rho<1$.**
>
> **A2** It appears that the reviewer confuses “superlinear convergence” with “quadratic convergence”. Specifically, the rate of $\rho^{2^t}$ mentioned by the reviewer is “quadratic convergence”, which is a special case of, but not equivalent to, “superlinear convergence”. In the optimization literature, the convergence is said to be Q-superlinear if $\lim_{t\rightarrow\infty}\frac{\\|x_{t+1}-x^\*\\| }{\\|x_t-x^\*\\|}=0$ (see, e.g., Appendix A.2 of Nocedal & Wright (2006)). Note that in Theorem 1 (c) and (d), we showed $\\|x_{t+1}-x^\*\\|=O(\frac{1}{t})\\|x_t-x^\*\\|$ and $\\|x_{t+1}-x^\*\\|=O(\frac{1}{\sqrt{t}})\\|x_t-x^\*\\|$, respectively. Since $\lim_{t \rightarrow \infty}\frac{1}{t}=\lim_{t\rightarrow\infty}\frac{1}{\sqrt{t}}=0$, these are superlinear convergence results by definition.
>
> ---
> **Q3 In Table 2, the iteration complexity of SNPE is $\log(\epsilon^{-1})$ similar to AGD.**
>
> **A3** Thank you for raising this point. Since our proposed SNPE method achieves superlinear convergence, it has a strictly better dependence on $\epsilon$ compared to AGD. In fact, in the iteration complexity of SNPE presented in Table 2, the dependence on $\epsilon$ can be replaced by $\frac{\log(\epsilon^{-1})}{\log(\log(\epsilon^{-1}))}$, which is provably better than the complexity of AGD by at least a factor of $\log\log(\epsilon^{-1})$.  We note that similar superlinear convergence rates have also been established in the prior work on stochastic Newton methods (Na et al., 2022) and in the literature on quasi-Newton methods (Rodomanov & Nesterov, 2022; Jin & Mokhtari, 2023). In our submission, we chose to use the simpler expression $O(\log(\epsilon^{-1}))$ to save space. However, we will update the table and include a discussion in the revision to more accurately reflect our superlinear convergence rate.
>
> ---
> **Q4 In lines 342-344, the dominating term in the complexity is the one that depends on $\epsilon$. The Damped Newton has better dependence than SNPE.**
>
> **A4**  We agree with the reviewer that damped Newton's method has a better dependence on $\epsilon$ than SNPE, which we also mentioned in lines 342-344. However, it is important to note that the damped Newton's method is deterministic and requires computing the exact Hessian, resulting in a per-iteration cost of $O(nd^2)$. In contrast, our method only requires a stochastic Hessian approximation and typically incurs a total per-iteration cost of $O(nd+d^3)$, as discussed in Section 6. This distinction is crucial because it allows us to achieve a better runtime compared to the damped Newton's method, especially when $n\gg d$. Thus, while the iteration complexity of damped Newton's method exhibits a better dependence on $\epsilon$, its overall arithmetic complexity can be much worse than ours. This is also demonstrated in our experiment presented in the shared PDF file. From Figures 1 and 2, we observe that while the damped Newton's method requires fewer iterations to converge, it takes more time overall to achieve the same accuracy as our method.
>
> ---
> **Q5 Empirical comparison between SNPE and damped Newton in terms of run-time.**
>
> **A5** Thank you for your suggestion. We have included the additional experiment in the shared PDF file; please see Figure 2. We would like to remark that damped Newton also performs a backtracking line search in every iteration, resulting in an overhead similar to ours. As expected, when the number of samples $n$ and the dimension $d$ are large, our method has a significant runtime gain compared to damped Newton, and the gap widens as the number of samples increases.
>
> ---
> **Q6 The authors ignore the complexity of line-search while computing the complexity of SNPE.**
>
> **A6** We respectfully disagree with the reviewer. As outlined in Remark 3 and proven in Appendix A.4, we explicitly characterize the cost of the line search in our SNPE method. Specifically, after $t$ iterations when $t = \tilde{\Omega}(\Upsilon^2/\kappa^2)$, the total number of line search steps can be bounded by $2t + \log(3M_1\sigma_0 /(\alpha \beta))$. Also, note that each line search step requires computing one gradient and one matrix inversion. Consequently, our method requires, on average, a constant number of gradient evaluations and matrix inversions per iteration. This leads to a constant overhead in the complexity bound, which is effectively hidden by the big O notation.
>
> ---
> **Additional References:**
>
> Boyd, S. and Vandenberghe L. Convex Optimization. Cambridge University Press, 2004.
>
> Nocedal, J. and Wright, S. J. Numerical Optimization. Springer, 2006.
>
> Rodomanov, A. and Nesterov, Y. Rates of superlinear convergence for classical quasi-Newton methods. Math. Program., 2022.
>
> Jin, Q. and Mokhtari, A. Non-asymptotic superlinear convergence of standard quasi-Newton methods. Math. Program., 2023.

---

> > ### Comment · Reviewer_BRp1 · 2024-08-09
> >
> > Thank you for your explanation of superlinear convergence and the plots.
> >
> > So SNPE performs better than Newton in the regime $n >> d$. Adding plots for other regimes to the appendix will be beneficial (just a suggestion to check the performance of SNPE compared to Newton).
> >
> > I will raise my score to 4.
> >
> > I am still not convinced about the iteration complexity in the table. You mentioned the iteration complexity of SNPE has dependence $\frac{\log(\varepsilon^{-1})}{\log \log(\varepsilon^{-1})}$. Is there a proof of this in the appendix that I can check? Else, can you add the proof here?

---

> > > ### Author Response · Authors · 2024-08-09
> > >
> > > Thank you for reading our rebuttal and for the follow-up comments.
> > >
> > > **SNPE v.s. Newton in the regime $n = O(d)$.**
> > >
> > > Thank you for the suggestion. We will include the additional plots for the regime where $n = O(d)$ in the appendix of our revision.
> > >
> > > ---
> > >
> > > **The iteration complexity of $\log(\epsilon^{-1})/\log\log(\epsilon^{-1})$.**
> > >
> > > Thank you for the question. Due to space constraints, we did not include the proof in the rebuttal. However, we will provide a proof sketch below.
> > >
> > > Note that for $t \geq \mathcal{U}\_3 = \tilde{O}(\Upsilon^2 + \kappa)$, we have $\\|x_{t+1}-x^\*\\| = \tilde{O}(\frac{\Upsilon}{\sqrt{t}}) \\|x_{t}-x^\*\\|$ by Theorem 2. By unrolling the inequality, this implies that $$\\|x_{t+1}-x^\*\\| \leq \\|x_{\mathcal{U}\_3}-x^\*\\| \tilde{O}\left(\prod_{s = \mathcal{U}\_3}^t \frac{\Upsilon}{\sqrt{s}}\right) \leq \\|x_0 - x^\*\\| \tilde{O}\left(\prod_{s = \mathcal{U}\_3}^t \frac{\Upsilon}{\sqrt{s}}\right).$$ Further, for $t \geq 2\mathcal{U}\_3$, we can upper bound $\frac{\Upsilon}{\sqrt{s}}$ as follows:
> > > - $\frac{\Upsilon}{\sqrt{s}} \leq \frac{\Upsilon}{\sqrt{\mathcal{U}\_3}} \leq 1$ for any $s \in [\mathcal{U}\_3, t/2]$,
> > > - $\frac{\Upsilon}{\sqrt{s}} \leq \frac{\Upsilon}{\sqrt{t/2}}$ for any $s \in [t/2,t]$.
> > >
> > > Thus, we have $$\prod_{s = \mathcal{U}\_3}^t \frac{\Upsilon}{\sqrt{s}} \leq \prod_{s = t/2}^t \frac{\Upsilon}{\sqrt{s}} \leq \left(\frac{\Upsilon}{\sqrt{t/2}}\right)^{\frac{t}{2}}.$$
> > >
> > > To derive a complexity bound, we upper bound the required number of iterations $t$ such that $ \left(\frac{\Upsilon}{\sqrt{t/2}}\right)^{\frac{t}{2}} = \epsilon$. Taking the logarithm of both sides and with some algebraic manipulation, we obtain $$\frac{t}{2\Upsilon^2} \log \frac{t}{2\Upsilon^2} = \frac{2}{\Upsilon^2} \log \frac{1}{\epsilon}.$$ Using the [Lambert W function](https://en.wikipedia.org/wiki/Lambert_W_function), the solution can be expressed as $ \log \frac{t}{2\Upsilon^2} = W(\frac{2}{\Upsilon^2} \log \frac{1}{\epsilon}) \Rightarrow t = 2\Upsilon^2 e^{ W(\frac{2}{\Upsilon^2} \log \frac{1}{\epsilon})}$. Finally, by applying the bound $e^{W(x)} \leq \frac{2x+1}{1 + \log (x+1)}$ for any $x \geq 0$, we conclude that $t = O\left(\frac{\log(\epsilon^{-1})}{\log(\log(\epsilon^{-1}))}\right)$. Note that in the above derivation, we ignore the additional logarithmic factor $\log (t)$ in our superlinear convergence rate. However, a more careful analysis will show that it does not affect the final complexity bound. We also refer the reviewer to a similar derivation in Appendix D.2 of Jiang et al. (2023), where the authors provide the same complexity bound for a similar convergence rate of $(1+ O(\sqrt{t}))^{-t}$.
> > >
> > > R. Jiang, Q. Jin, and A. Mokhtari. "Online learning guided curvature approximation: A quasi-Newton method with global non-asymptotic superlinear convergence." COLT 2023.

---

> > > > ### Comment · Reviewer_BRp1 · 2024-08-09
> > > >
> > > > Thank you for the details. Please add them in the updated version.
> > > >
> > > > I will raise my score to 5.

---

### Official Review · Reviewer_m5uq · 2024-07-12

**Soundness:** 3
**Presentation:** 3
**Contribution:** 3
**Rating:** 6
**Confidence:** 4

**Summary:**

This paper uses the hybrid proximal extragradient framework to accelerate the convergence of Hessian average. The theoretical results significantly reduce the number of iterations to enter the linear phase, initial superlinear phase, and final superlinear phase when compared to the initial Hessian average method.

**Strengths:**

The theoretical results of this paper are impressive. It improves the results of Hessian average [Na e.t.al 2022]. The idea of incorporating NPE framework and Hessian average is interesting. The paper is generally well-written and the results are easy to follow.

**Weaknesses:**

This paper does not provide empirical results for the proposed methods against AGD, which is necessary.
The proposed methods still require exact gradient oracle and its iteration complexity is $O({\kappa}+\log(1/\epsilon))$, while the classical AGD method require only $O(\sqrt{\kappa}\log(1/\epsilon))$. It is very important to use numerical results to exhibit the benefits of using second-order information.

**Questions:**

Refer to weakness part.

---

> ### Author Rebuttal · Authors · 2024-08-06
>
> We thank the reviewer for their positive feedback.
>
> **Q1 Empirical comparison with AGD.**
>
> **A1** Following your suggestion, we compared the performance of our SNPE method against AGD in our new experiment; please see Figures 1 and 2 in the shared PDF file. From Figure 1, we observe that our SNPE method, with either uniform or weighted averaging, requires far fewer iterations to converge than AGD due to the use of second-order information. Consequently, while SNPE has a higher per-iteration cost than AGD,  it converges faster overall in terms of runtime, as demonstrated in Figure 2.

---

> > ### Comment · Reviewer_m5uq · 2024-08-10
> >
> > I thank the authors for their response and have no further questions.

---

### Official Review · Reviewer_qDEj · 2024-07-31

**Soundness:** 3
**Presentation:** 4
**Contribution:** 3
**Rating:** 7
**Confidence:** 2

**Summary:**

Newton method is well-known for its local quadratic convergence. However, the use of Hessian introduces additional computation challenges. One approach to tackling this issue is an inexact approximation of the Hessian. In this paper, the authors consider the finite sum minimization problem. They propose a stochastic approximation of the Newton method, where instead of the full Hessian they use its subsample approximation. The idea of the proposed method is based on another known approach, that uses weighted average over a subsample of Hessians. In this paper, the authors improve this approach using the Hybrid Proximal Extragradient (HPE) framework for strongly convex problems. As a result, the proposed algorithm achieves both linear and superlinear convergence areas in fewer iterations by improving the dependence on condition number of convergence rate.

**Strengths:**

This paper introduces a new type of stochastic inexact Newton method with a better convergence rate than existing analogs. The paper is written in a clear way, sketchily describing its main points, whereas most of the technical details are provided in the appendix. As for me, this is a good way to describe your idea in such a limited space. Authors provide an intuition or explanation after every lemma and theorem, which is also a good practice.

**Weaknesses:**

Authors consider only a strongly convex setup. However they mention extending their approach to convex case, this seems to me as a major limitation of this work. Additionally, to check condition (3) authors employ line-search, which introduces an additional logarithmic factor in the convergence rate. However this can be the burden of the HPE framework, it seems not very significant, but still an issue.

## Minor remarks
1. Line 168: remove "follows" from the end
2. Line 168: not Step 5, but Step 6
3. Sometimes you write "stepsize", sometimes - "step size". Please, be consistent.

**Questions:**

1. Why do you use $\sigma_{t+1} = \eta_t/\beta$? Please provide more details.

**Limitations:**

No limitations.

---

> ### Author Rebuttal · Authors · 2024-08-06
>
> **Q1 Authors consider only a strongly convex setup and this seems to me as a major limitation of this work.**
>
> **A1** Thank you for raising this point. To begin with, we note that the strong convexity assumption is common in the study of stochastic second-order methods, including Subsampled Newton (Erdogdu \& Montanari, 2015; Roosta-Khorasani \& Mahoney, 2019), Newton Sketch (Pilanci \& Wainwright, 2017; Agarwal et al., 2017), and notably, the recent work by Na et al. (2022). Hence, by establishing our results within the strongly convex setting, we can better position our contribution in relation to prior work.
>
> Moreover, the focus on strongly convex functions in these works, as well as in ours, stems from the clear advantages that stochastic second-order methods offer over first-order methods, such as gradient descent. Specifically, stochastic second-order methods achieve a superlinear convergence rate, as demonstrated in this paper, which is superior to the linear rate attained by first-order methods. We also note that the assumption of strong convexity is necessary for achieving superlinear convergence rates, as only a sublinear rate can be achieved in the convex setting, even with exact Hessian information.
>
> Finally, while we believe it is possible to extend our techniques to the convex setting, developing the necessary theory and discussing the results would require more space than is available in the submission. Therefore, this extension is beyond the scope of this paper.
>
> ---
>
> **Q2 To check condition (3) authors employ line-search, which introduces an additional logarithmic factor in the convergence rate. However, this can be the burden of the HPE framework, it seems not very significant, but still an issue.**
>
> **A2**
> The reviewer is correct in noting that we need to employ line search to ensure Condition (3). However, we would like to mention that most stochastic second-order methods require some form of line search to ensure global convergence, and this limitation is not unique to our methods. For instance, Pilanci \& Wainwright (2017), Roosta-Khorasani \& Mahoney (2019), and Na et al. (2022) all used a backtracking line search to ensure a sufficient decrease condition.
>
> Moreover, we wish to clarify that the line search scheme only introduces an **additive** logarithmic factor, instead of a multiplicative one, in our final complexity bound. Specifically, as mentioned in Remark 3 and proved in Appendix A.4, after $t$ iterations when $t = \tilde{\Omega}(\Upsilon^2/\kappa^2)$, the total number of line search steps can be bounded by $2t + \log(3M_1 \sigma_0 /(\alpha \beta))$. Also, note that each line search step requires computing one gradient and one matrix inversion. Consequently, our method requires, on average, a constant number of gradient evaluations and matrix inversions per iteration, leading to a constant overhead in the complexity bound.
>
> ---
>
> **Q3 Why do you use $\sigma_{t+1} = \eta_t/ \beta$?**
>
> **A3**
> This is a good question. Our motivation behind the choice $\sigma_{t+1} = \eta_t/ \beta$ is to allow the step size to grow, which is necessary for achieving a superlinear convergence rate. Specifically, our entire convergence analysis builds on Proposition 1, which demonstrates that $\\|x_{t+1}-x^\*\\|^2 \leq \\|x_{t}-x^\*\\|^2 (1+2\eta_t \mu)^{-1}$. Consequently, we require the step size $\eta_t$ to go to infinity to ensure that $\lim_{t \rightarrow \infty} \frac{\\|x_{t+1}-x^\*\\|}{\\|x_{t}-x^\*\\|} = 0$.  Note that this would not be possible if we simply set $\sigma_{t+1} = \eta_t$, since it would automatically result in $\eta_{t+1} \leq \sigma_{t+1} \leq \eta_t$. Moreover, this condition $\sigma_{t+1} = \eta_{t}/\beta$ is explicitly utilized in Lemmas 8 and 16 in the appendix, where we demonstrate that  $\eta_t$ can be lower bounded by the minimum of $\sigma_0/\beta^t$ and another term.
>
> We should note that this more aggressive choice of the initial step size at each round could potentially increase the number of backtracking steps. However, as mentioned in the response to **Q2** above, this does not cause a significant issue, since the average number of backtracking steps per iteration can be bounded by a constant close to 2.
>
> Thank you again for the question and we will include the discussions above in our revision.
>
> ---
>
> **Q4 Minor remarks.**
>
> **A4** Thank you for catching the typos. We will fix them all in the revision.
>
> -----
> **References:**
>
> Erdogdu, M. A. and Montanari, A. Convergence rates of sub-sampled newton methods. Advances in Neural Information Processing Systems, 2015.
>
> Roosta-Khorasani, F. and Mahoney, M. W. Sub-sampled Newton methods. Mathematical Programming, 2019.
>
> Pilanci, M. and Wainwright, M. J. Newton sketch: A near
> linear-time optimization algorithm with linear-quadratic
> convergence. SIAM Journal on Optimization, 2017.
>
> Agarwal, N., Bullins, B., and Hazan, E. Second-order
> stochastic optimization for machine learning in linear
> time. The Journal of Machine Learning Research, 2017.
>
> Na, S., Derezinski, M., and Mahoney, M. W. Hessian averaging in stochastic Newton methods achieves superlinear convergence. Mathematical Programming, 2022.

---

> > ### Comment · Reviewer_qDEj · 2024-08-14
> >
> > I thank the authors for the answers. I don't have any further questions and increased my rating to 7.

---

### Author Rebuttal · Authors · 2024-08-06

We thank the reviewers for their insightful feedback. Overall, Reviewers **qDEj**, **m5uq** and **TogL** provided largely positive comments, highlighting the strength of our theoretical analysis and the clarity of the presentation. Reviewer **BRp1** raised some concerns regarding our complexity results, which we have addressed in detail by clarifying our superlinear convergence rates and discussing the complexity of line search.

Following your suggestions, we have performed a new set of experiments, and the plots are included in the shared PDF file.
We considered minimizing the regularized log-sum-exp function as in our submission, where the regularization parameter $\lambda$ is $10^{-3}$, the dimension $d$ is 500, and the number of samples $n$ is chosen from 50,000, 10,000, and 150,000, respectively. Note that we increased the dimension and the number of samples to better demonstrate the efficiency of our method in a large-scale setting.

For Figures 1 and 2, we implemented a variant of SNPE without the extragradient step (see the discussions in Section 7) and compared it against the stochastic Newton method in [1], accelerated gradient descent (AGD), damped Newton's method, and Newton Proximal Extragradient (NPE, i.e., our SNPE method with exact Hessian). For the stochastic Hessian estimate, we use a subsampling strategy with a subsampling size of $s = d = 500$. Moreover, in Figure 3, we compare the two variants of SNPE: one with the extragradient step (as described in Algorithm 1) and one without (used in the previous plots).

- **Comparison with AGD.** From Figure 1, we observe that our SNPE method, with either uniform or weighted averaging, requires far fewer iterations to converge than AGD due to the use of second-order information. Consequently, while SNPE has a higher per-iteration cost than AGD,  it converges faster overall in terms of runtime, as demonstrated in Figure 2.

- **Comparison with the damped Newton's method and NPE**. As expected, since both damped Newton and NPE use exact Hessian, Figure 1 shows that they exhibit superlinear convergence and converge in fewer iterations than the other algorithms. However, since the exact Hessian matrix is expensive to compute, they incur a high per-iteration computational cost and overall take more time than our proposed SNPE method to converge (see Figure 2). Moreover, the gap between these two methods and SNPE widens as the number of samples $n$ increases, demonstrating the advantage of our method in the large data regime.

- **Effect of the extragradient step**. In Figure 3, we test the effect of the extragradient step in our proposed SNPE method. We observe that in all cases, the variant without an extragradient step outperforms the original version, suggesting that the extragradient step may not be beneficial for minimization problems. Nevertheless, the SNPE method with the extragradient step, which is the one analyzed in our paper, still outperforms the stochastic Newton method in [1].

We will revise our submission to include the new figures and discussions.

---

### Comment · Area_Chair_LBpu · 2024-08-11
**Discussion**

Dear Reviewers,

Thank you for your effort in reviewing this paper. Please go over the authors rebuttal (and I suggest also other reviews and rebuttals to those) and see if it changes your mind or answers any of your concerns. In particular please engage in discussion with authors on points which have been clarified in the rebuttal or discuss why the rebuttal does not answer your concerns.

Thanks you!
AC

---

### Decision · Program_Chairs · 2024-09-25

**Decision:**

Accept (poster)

**Comment:**

Overall, all reviewers were positive about this work and I support their decision.